# LOGIC-LOGIT: A LOGIC-BASED APPROACH TO CHOICE MODELING

**Shuhan Zhang, Wendi Ren & Shuang Li** [*]
CUHK-Shenzhen
{shuhanzhang,wendiren}@link.cuhk.edu.cn,lishuang@cuhk.edu.cn

## ABSTRACT

In this study, we propose a novel rule-based interpretable choice model, **Logic-Logit**, designed to effectively learn and explain human choices. Choice models have been widely applied across various domains—such as commercial demand forecasting, recommendation systems, and consumer behavior analysis—typically categorized as parametric, nonparametric, or deep network-based. While recent innovations have favored neural network approaches for their computational power, these flexible models often involve large parameter sets and lack interpretability, limiting their effectiveness in contexts where transparency is essential.

Previous empirical evidence shows that individuals usually use *heuristic decision rules* to form their consideration sets, from which they then choose. These rules are often represented as *disjunctions of conjunctions* (i.e., OR-of-ANDs). These rules-driven, *consider-then-choose* decision processes enable people to quickly screen numerous alternatives while reducing cognitive and search costs. Motivated by this insight, our approach leverages logic rules to elucidate human choices, providing a fresh perspective on preference modeling. We introduce a unique combination of column generation techniques and the Frank-Wolfe algorithm to facilitate efficient rule extraction for preference modeling—a process recognized as NP-hard. Our empirical evaluation, conducted on both synthetic datasets and real-world data from commercial and healthcare domains, demonstrates that Logic-Logit significantly outperforms baseline models in terms of interpretability and accuracy.

## 1 INTRODUCTION

In today's complex decision-making environment, individuals usually encounter an overwhelming array of options to choose from. In high-stakes areas like healthcare, patients face critical decisions such as selecting healthcare providers, hospitals, and insurance plans—choices that directly influence their health outcomes and financial well-being. Similarly, doctors must navigate a multitude of medications and treatment plans to determine the best course of action for their patients. In the realm of consumer behavior, homebuyers are tasked with evaluating properties based on various factors, such as location, price, and amenities, all of which contribute to their decision-making processes. This abundance of choices not only highlights the daily challenges individuals face but also underscores the need for models that can effectively capture the heterogeneous and intricate patterns inherent in their decision-making.

Empirical research reveals that individuals often simplify decision-making through a *consider-then-choose* process (Hauser, 2014). In this cognitive framework, people first establish a consideration set using heuristic rules—*logical* guidelines that filter options before selecting from this reduced set. These rules, often informed by past experiences and contextual knowledge, are typically represented as *disjunctions of conjunctions* (OR-of-ANDs). This logical structure enables individuals to efficiently screen alternatives, manage the complexity of decision-making, and reduce cognitive effort and search costs. Understanding and modeling this structured decision-making process is critical for developing systems that can predict and explain human choices.

Inspired by these insights, we propose the **Logic-Logit model**, a novel rule-based choice model designed to capture and explain heterogeneous human choice patterns through OR-of-ANDs logic rules. Our approach addresses two fundamental challenges: *1) representing diverse human deci-*

---

[*]Corresponding author

*sion rationales using OR-of-ANDs logic rules*, and *2) efficiently extracting compact rule sets from preference datasets despite the inherent combinatorial complexity.*

In our framework, human preferences are modeled through a *rule-based utility function*, capturing decision-making via disjunctions of conjunctions (OR-of-ANDs) logic rules. These logic-informed features, grounded in data, provide a structured and interpretable representation of how individuals evaluate and prioritize options. The utility of each option is modeled as a linear combination of these logic-based features, ensuring that the model remains interpretable while capturing the complexity and heterogeneity of human preferences, where different preference types may utilize various rules with distinct weights. To estimate human preference types and their associated rule sets, we employ a two-tiered optimization framework:

In the **outer loop**, we utilize the Frank-Wolfe algorithm (conditional gradient) to systematically search for new preference types and estimate their proportions. This algorithm is well-suited for constrained convex optimization, enabling efficient exploration of the model space by iteratively updating preference types based on their respective logic rules (Jagabathula et al., 2020). Continuous refinement through the Frank-Wolfe algorithm ensures the model captures diverse decision-making behaviors while optimizing overall structure.

In the **inner loop**, the column generation algorithm effectively addresses the combinatorial nature of rule search by iteratively refining the rule set (Wei et al., 2019). This approach alternates between two essential tasks: solving a master problem to update the weights of the existing rules and tackling a subproblem to identify new rules that can enhance model performance. We begin with a small, manageable rule set and progressively introduce new rules as the algorithm iterates. The subproblem employs a risk-seeking search strategy, focusing on rules with the greatest potential for performance gains, which encourages a more exploratory approach. This ensures that the rule set remains diverse and prevents premature convergence to suboptimal solutions.

Our contributions can be summarized as follows:

1) We introduce the Logic-Logit model, which effectively combines interpretability with computational efficiency, providing a structured framework for capturing human choice patterns.

2) By utilizing OR-of-ANDs logic rules and integrating the Frank-Wolfe algorithm with column generation, we address the complexities involved in understanding human decision-making logic and rule extraction.

3) Our empirical evaluation, conducted on synthetic datasets and real-world data from commercial and healthcare domains, demonstrates that the Logic-Logit model significantly outperforms baseline models in interpretability and accuracy, effectively bridging the gap between identifying heterogeneity and clarifying underlying decision-making logic.

## 2 RELATED WORKS

### 2.1 CHOICE MODEL AND PREFERENCE LEARNING

**Choice models** provide structured frameworks for examining how individuals assess various alternatives during the decision-making process. This constitutes a pivotal element in preference learning. Guided by the Random Utility Theory (RUT) proposed by McFadden (1981), the primary objective of choice modeling is to establish a mapping function that translates option features into a utility value, which is a real number.

The Multinomial Logit (MNL) Model, as introduced by Hausman & McFadden (1984), serves as the cornerstone for nearly all extant models and has been diversified into numerous extensions to accommodate various scenarios. For instance, the Nested Logit Model, developed by Wen & Koppelman (2001) and Hensher & Greene (2002), addresses issues of independence among options. The Mixture MNL, as explored by Jagabathula et al. (2020), Bhat (1997), and Greene & Hensher (2003), is designed to handle complex distributions within the mapping function. Both our model and the model proposed by Jagabathula et al. (2020) are grounded in the Frank-Wolfe framework; however, the utility value of options in his model is computed using a linear approach.

More recently, Neural Network-based MNL models, such as those presented by Aouad & Désir (2022) and Wang et al. (2020), have been introduced to capture intricate and non-linear relationships between utility values and option features. These models leverage the power of neural networks to enhance the predictive accuracy and flexibility of traditional MNL models, addressing the limitations of linear assumptions and enabling more sophisticated modeling of choice behaviors.

In the above work, all alternatives are evaluated simultaneously. In contrast, Consider-then-Choose models adopt a two-stage decision process: first, consumers form a consideration set using screening rules or heuristics; second, they select the option that maximizes utility within the consideration set. Liu & Arora (2011) highlight the conjunctive screening rule, where a product is considered only if all its features meet specific criteria. Aouad et al. (2021) develop a dynamic programming framework to analyze the computational aspects of assortment optimization under this paradigm. Akchen & Mitrofanov (2023) emphasize the importance of consideration sets, defining a nonparametric choice model characterized solely by a distribution over these sets.

## 2.2 Linear Rule Learning

**Logic Rule**   Rule learning has long been an important technique in supervised learning, particularly in fields that demand model interpretability. As indicated by Rivest (1987) and Wei et al. (2019), Logic Rules are conceptualized as the process of thresholding original option features into binary features. These rules can take the form of conjunctions represented by the logical operator $\wedge$.

**Rule Learning**   The number of potential rules that can be derived from thresholding and conjunctions is virtually infinite, making the identification of appropriate rule sets and their corresponding weights in influencing choice behavior an NP-hard problem. Various methods have been proposed to tackle this challenge. Ruczinski et al. (2003) employed multiple logic trees fitted to the data, utilizing regression on the weights of these trees; however, the adjustment of logic trees proved to be time-consuming.

To address the inefficiencies associated with traditional approaches, Wei et al. (2019) introduced Generalized Linear Rule Models and implemented a column generation algorithm—an advanced technique for optimizing problems afflicted by dimensionality issues (Desaulniers et al., 2006). This approach involves an iterative process of searching for and evaluating potential rules, thereby enabling the refinement of the rule set through reweighting. Chen & Mišić (2022) proposed a decision tree-based choice model that incorporates column generation to capture the mutual effects among options. However, the applicability of this model is limited by its inability to leverage option-specific features and the interpretability deficit inherent in random forests, whereas rules can be readily understood and explicated.

## 3 Background: Discrete Choice Model

Discrete choice models describe individuals' preferences among a set of alternatives. Let $S$ be a set of alternatives, each characterized by a feature vector $\boldsymbol{x}_s$, $s \in S$. Let $\boldsymbol{x} = [\boldsymbol{x}_s]_{s \in S}$. Given a choice set, it predicts the probability that an individual select one of the alternatives. The choice probability of an alternative $s$ is based on its utility, which is a function of the feature vector. For example, one can consider a linear utility function, $f(\boldsymbol{x}_s) = \boldsymbol{w}^\top \boldsymbol{x}_s$, where $\boldsymbol{w}$ is the taste vector to be estimated.

**Multinomial Logit**   This is the most classic choice model traced back to Plackett (1975); Luce (1959). With a linear utility, it models the choice probability as a softmax function

$$\frac{\exp\left(\boldsymbol{w}^T \boldsymbol{x}_s\right)}{\sum_{s' \in S} \exp\left(\boldsymbol{w}^T \boldsymbol{x}_{s'}\right)}, \quad s \in S.$$

This model can be explained through Random Utility Maximization theory (McFadden, 1981). In this framework, the utility of item $s$ is random, expressed as $U_s = \boldsymbol{w}^T \boldsymbol{x}_s + \epsilon_s$, where $\epsilon_s$ is the Gumbel noise. When the decision maker chooses the item that maximizes the random utility, i.e., $\arg\max_{s \in S} U_s$, the resulting choice probability is given by the softmax expression above.

**Mixed Logit**   To account for heterogeneity in the taste vector $\boldsymbol{w}$, the mixed logit model assumes that $\boldsymbol{w}$ is drawn from a latent distribution $\rho$, referred to as the mixture distribution. Each user behaves according to a multinomial logit, but their specific taste vector is unobservable and treated as a random sample from the population distribution $\rho$. Consequently, the choice probability is

$$\mathbb{E}_{\boldsymbol{w}} \sim \rho \left[ \frac{\exp\left(\boldsymbol{w}^T \boldsymbol{x}_s\right)}{\sum_{s' \in S} \exp\left(\boldsymbol{w}^T \boldsymbol{x}_{s'}\right)} \right], \quad s \in S.$$

# 4 OUR PROPOSED MODEL: LOGIC-LOGIT

## 4.1 MOTIVATING EXAMPLE

Let's take house-buyers as an example. Different buyer types have unique house-purchasing preferences, represented by human-readable OR-of-ANDs logic rules for filtering unsuitable options. Middle-class families with children prioritize homes meeting criteria like $A$ (price below $\$500,000$), $B$ (within 5 miles of a good school), or $C$ (within 10 miles of work). Their choice rule is $(A \wedge B) \vee (A \wedge C)$, preferring affordable homes near a good school or work. Single professionals focus on criteria such as $D$ (modern amenities), $E$ (within 3 miles of entertainment), or $F$ (within 5 miles of work). Their rule is $(D \wedge E) \vee (D \wedge F)$, favoring homes with modern features and close to entertainment or work. Wealthy buyers look for $G$ (luxury features), $H$ (exclusive neighborhood), or $I$ (large lot size). Their choice logic is $(G \wedge H) \vee (G \wedge I) \vee (H \wedge I)$, preferring luxury properties in an exclusive area or with a large lot.

These logic - based preferences provide a structured way to capture diverse customer needs. In our model, each customer type has its own set of rules.

## 4.2 LOGIC-INFORMED UTILITY AND LOGIC-LOGIT

For each customer type, e.g., middle-class families with children or single professionals, each has its own distinct OR-of-ANDs logic rule to determine their preferences. We define a set of predicates $\mathcal{P} = \{p_m\}_{m=1}^M$, where each predicate $p_m : \mathcal{X} \to \{0,1\}$ maps feature vectors, $\boldsymbol{x} \in \mathcal{X}$ to Boolean values $\mathcal{X}$ refers to the set of all choice features. A predicate evaluates to 1 if the condition it specifies is satisfied and 0 otherwise. These mappings, from real-valued features to predicates, are predefined and fixed. A possible way of predefining the $p_m$ and $\mathcal{P}$ is shown in Appendix A, which is also used in our real data experiments.

Given these predicates, an OR-of-ANDs logic rule can be expressed as

$$R = \bigvee_{u=1}^U \Big( \bigwedge_{p_m \in \mathcal{A}_u} p_m \Big)$$

where $R$ represents the disjunction of $U$ conjunction rules specific to a customer type. This is a general form that can express any logic rules. Each conjunction consists of predicates in the set $\mathcal{A}_u$, corresponding to the $u$-th conjunction for that particular type. Given $M$ predicates, there are $2^M - 1$ possible different conjunctions, each of which corresponds to a non-empty subset of $\mathcal{P}$, the set of all predicates. The evaluation of the rule $R$ for an instance $\boldsymbol{x}$ is $R(\boldsymbol{x}) = \bigvee_{u=1}^U \Big( \bigwedge_{p_m \in \mathcal{A}_u} p_m(\boldsymbol{x}) \Big)$.

For our proposed Logic-Logit model, the utility function for the customer is modeled as a weighted combination of these conjunctions, with each conjunction acting as a Boolean feature:

$$\sum_{u=1}^U w_u \cdot \Big( \bigwedge_{p_m \in \mathcal{A}_u} p_m(\boldsymbol{x}) \Big),$$

where $w_u$ is the weight associated with the $u$-th conjunction. Under the multinomial logit model, the probability of selecting item $s$ from the offer set $S$ can then be expressed as

$$P_{\boldsymbol{w}}(s \mid S, \boldsymbol{x}) = \frac{\exp\Big( \sum_{u=1}^U w_u \cdot \big( \bigwedge_{p_m \in \mathcal{A}_u} p_m(\boldsymbol{x}_s) \big) \Big)}{\sum_{s' \in S} \exp\Big( \sum_{u=1}^U w_u \cdot \big( \bigwedge_{p_m \in \mathcal{A}_u} p_m(\boldsymbol{x}_{s'}) \big) \Big)}, \quad s \in S. \tag{1}$$

We remark that the dimension of $\boldsymbol{w}$ is $2^M - 1$, which corresponds to the number of all possible Boolean features generated from conjunction rules derived from the predicate set $\mathcal{P}$. On the other hand, it is a sparse vector with only $U$ nonzero elements $\{w_u\}_{u \in [U]}$.

## 4.3 MIXED LOGIC-LOGIT MODEL

To account for the diversity in user preferences, we extend the above-introduced Logic-Logit model to a Mixed Logic-Logit model. In this case, the rule weight vector $\boldsymbol{w}$ is modeled as a sample drawn from a latent mixture distribution $\rho(\boldsymbol{w})$. The choice probability of selecting item $j$ from the offer set $S, \boldsymbol{x}$ under this mixture distribution is

$$P_\rho(s \mid S, \boldsymbol{x}) = \mathbb{E}_{\boldsymbol{w} \sim \rho}[P_{\boldsymbol{w}}(s \mid S, \boldsymbol{x})], \tag{2}$$

where $P_{\boldsymbol{w}}(s \mid S, \boldsymbol{x})$ is computed in Eq. (1). We denote the mixed logit choice probability vector from a mixture distribution $\rho$ as

$$P_\rho(S, \boldsymbol{x}) = [P_\rho(s \mid S, \boldsymbol{x})]_{s \in S}.$$

## 5 LEARNING ALGORITHM

Let $\mathcal{D}$ be the data distribution, with each data point $(S, \boldsymbol{x}, y)$ records the choice set $S$, their feature vectors $\boldsymbol{x}$, and the corresponding choice frequency $y$, which is either a probability vector or a one-hot vector. To learn the mixture distribution $\rho$, consider minimizing the negative log-likelihood

$$\min_{\rho \in \mathcal{P}} \ L(\rho) := -\mathbb{E}_{(S,\boldsymbol{x},y) \sim \mathcal{D}} \left[ y^\top \log P_\rho(S, \boldsymbol{x}) \right]. \tag{3}$$

This problem is an infinite-dimensional functional optimization problem over the mixture distribution $\rho$. In addition, from the discussion in Section 4.3, the taste vector $\boldsymbol{w}$ is potentially of high dimensions due to exponentially many potential rules. On the other hand, in many practical situations, only a small number of simple rules serve as the main driving force of a decision-making process. To facilitate an interpretable way to generate sparse mixture distributions, we adopt a functional conditional gradient algorithm to solve (3). The algorithm's road map is illustrated in Fig. 1.

### 5.1 FUNCTIONAL CONDITIONAL GRADIENT ALGORITHM FOR ESTIMATING THE MIXING DISTRIBUTION

We perform a functional conditional gradient (a.k.a. Frank-Wolfe) algorithm (Bach, 2017) to solve the problem Eq. (3). To this end, we first compute the functional gradient of $L$ at $\rho$, which is itself a function, as

$$L'(\rho)(\boldsymbol{w}) = -\mathbb{E}_{(S,\boldsymbol{x},y) \sim \mathcal{D}} \left[ \sum_{s \in S} \frac{y_s}{P_\rho(s \mid S, \boldsymbol{x})} \cdot P_{\boldsymbol{w}}(s \mid S, \boldsymbol{x}) \right]. \tag{4}$$

The core of the conditional gradient descent algorithm is the Frank-Wolfe step. At each iteration $k$, given a feasible solution $\rho_k$, this step computes the optimal solution $\bar{\rho}_k$ by solving the following optimization problem based on the gradient $L'(\rho_k)$:

$$\bar{\rho}_k \in \arg\min_{\rho \in \mathcal{P}} \langle L'(\rho_k), \rho \rangle,$$

where the inner product is between a function and a distribution, i.e., $\langle L'(\rho_k), \rho \rangle = \mathbb{E}_\rho[L'(\rho_k)]$. Using the functional gradient (4), the above minimization problem can be rewritten as

$$\min_{\rho \in \mathcal{P}} \ -\mathbb{E}_{(S,\boldsymbol{x},y) \sim \mathcal{D}} \left[ \sum_{s \in S} \frac{y_s}{P_{\rho_k}(s \mid S, \boldsymbol{x})} \cdot \mathbb{E}_\rho \big[ P_{\boldsymbol{w}}(s \mid S, \boldsymbol{x}) \big] \right].$$

Thanks to the linearity of the objective in $\rho$, it suffices to restrict to Dirac measures, i.e., $\rho = \delta_{\boldsymbol{w}}$ for some taste vector $\boldsymbol{w}$. This leads to the following equivalent form of the Frank-Wolfe step:

$$\min_{\boldsymbol{w}} \ -\mathbb{E}_{(S,\boldsymbol{x},y) \sim \mathcal{D}} \left[ \sum_{s \in S} \frac{y_s}{P_{\rho_k}(s \mid S, \boldsymbol{x})} \cdot P_{\boldsymbol{w}}(s \mid S, \boldsymbol{x}) \right]. \tag{FW}$$

Therefore, the Frank-Wolfe step finds a new customer type, denoted as $\boldsymbol{w}_k$. Given $\delta_{\boldsymbol{w}_k}$, the next iteration $\rho_{k+1}$ is defined as

$$\rho_{k+1} \leftarrow (1 - \eta_k)\rho_k + \eta_k \delta_{\boldsymbol{w}_k},$$

where $\eta_k$ is a suitable step size parameter, often chosen as $\rho_k = \frac{2}{k+1}$ to achieve a sublinear convergence rate $O(1/K)$, or determined by line search.

Practically, one can start with a single customer type $\rho_0 = \delta_{\boldsymbol{w}_0}$, where $\boldsymbol{w}_0$ is an initial parameter vector, such as the best multinomial logit weights. The algorithm then iteratively adds new customer types to the mixture distribution by solving the Frank-Wolfe step (FW). To achieve better empirical performance, one can also perform a fully corrective step, which updates the mixture distribution by re-optimizing the weights on all customer types:

$$\rho_{k+1} \leftarrow \arg\min_\rho \left\{ L(\rho) : \ \rho = \sum_{j=0}^k \alpha_j \delta_{\boldsymbol{w}_j}, \alpha \in \Delta^K \right\}.$$

This is a convex optimization problem over $\alpha$ and can be solved efficiently.

On a high level, this algorithm shares similar aspects as the finite-dimensional conditional gradient algorithms in Jagabathula et al. (2020); Hu et al. (2022), which estimate mixed logit model by re-parameterizing it as a constrained convex optimization over choice probability vectors. Different

from theirs, we develop our algorithm using the functional conditional gradient. Moreover, it is important to note that in our logic-rule-based problem, the Frank-Wolfe step (FW) has a different computational challenge than they have, as ours involves searching over a space with a combinatorial dimension $2^M - 1$. We devote the next section to discussing how to solve this problem efficiently.

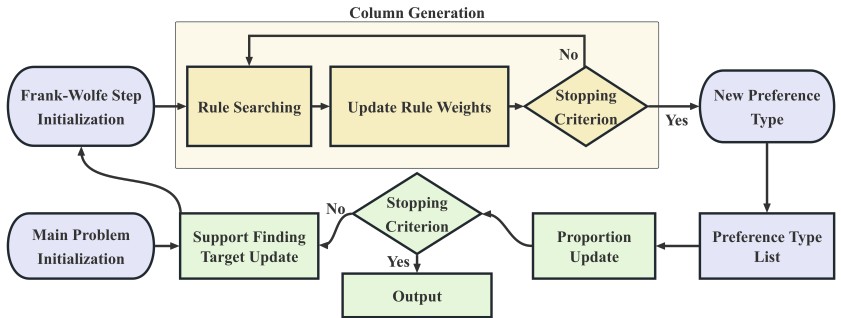

Figure 1: The framework of our Algorithm.

## 5.2 SOLVING FRANK-WOLFE STEP USING COLUMN GENERATION

Plugging in our logic-logit model (1) to the Frank-Wolfe step (FW) yields the following objective

$$\phi(\boldsymbol{w} \mid \{\mathcal{A}_u\}_{u=1}^U) := \tag{5}$$

$$- \mathbb{E}_{(S,\boldsymbol{x},y)\sim\mathcal{D}} \left[ \sum_{s\in S} \frac{y_s}{P_{\rho_k}(s \mid S, \boldsymbol{x})} \cdot \frac{\exp\left(\sum_{u=1}^U w_u \cdot \left(\bigwedge_{p_m\in\mathcal{A}_u} p_m(\boldsymbol{x}_s)\right)\right)}{\sum_{s\in S} \exp\left(\sum_{u=1}^U w_u \cdot \left(\bigwedge_{p_m\in\mathcal{A}_u} p_m(\boldsymbol{x}_{s'})\right)\right)} \right]. \tag{6}$$

When the number of predicates $M$ is large, it results in an exponentially large number of possible conjunctions. To address this challenge, we devise a column generation algorithm. This approach begins with a smaller, more manageable problem using a limited search space, and then gradually expands the space by adding new rules iteratively. We start with an empty rule set: $R^{(0)} = \{\}$, and initialize rule weights $\boldsymbol{w}^{(0)} = []$. We will repeat the following procedures until the convergence or the stopping criterion is met.

**Master Problem: Update Rule Weights** Suppose this current rule set is $R^{(U)} = \{\mathcal{A}_u\}_{u=1}^U$, the master problem optimizes the weights by minimizing the Frank-Wolfe objective

$$\boldsymbol{w}^{(U)} \leftarrow \arg\min_{\boldsymbol{w}} \phi\left(\boldsymbol{w} \mid \{\mathcal{A}_u\}_{u=1}^U\right).$$

This step adjusts the weights for the current set of rules to best fit the data by minimizing the Frank-Wolfe objective $\phi$. Note that this optimization is on the $U$-dimensional space and thus is more tractable.

**Subproblem: Search for a New Rule** With the rule weights $\boldsymbol{w}^{(U)}$ fixed, the subproblem focuses on identifying a new rule $\mathcal{A}_{U+1}$ that can further enhance the objective. Specifically, if the gradient vector $\nabla\phi(\boldsymbol{w}^{(U)}) \in \mathbb{R}^{2^M-1}$ is not a zero vector, then there exists a descent direction that reduces the objective (5). To identify a non-zero component of this high-dimensional gradient vector, we conduct a rule search as follows.

The conjunction rule is identified as a path in a tree, where each node represents a predicate. We employ two search strategies: Breadth-First Search (BFS) and Depth-First Search (DFS). BFS first examines all rules with the same length and then gradually increases rule length; whereas DFS focuses on generating new rules by performing conjunctions on the rules that have already been identified. To streamline the process, we impose a limit on rule length to ensure that we focus on rules that are both computationally feasible and manageable. Moreover, we take a risk-seeking approach by prioritizing the maximization of potential gains over average outcomes. Namely, when comparing the performance of different rules, we focus on specific quantiles of the performance distribution rather than the average. This allows us to identify rules that could yield higher rewards or provide valuable insights. Such an exploratory strategy is especially useful empirically, where certain rules might produce substantial gains in specific scenarios but could be overlooked when

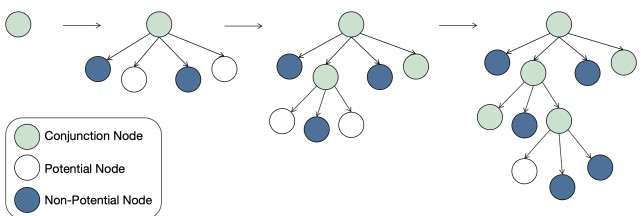

Figure 2: *DFS Visualization* In the figure, each node corresponds to a predicate. When the search rule conjunction length increases from $n$ to $n + 1$, the depth-first search (DFS) algorithm initiates from the previously detected $n$-conjunction rules and exhaustively searches through all extensions with other predicates in order to discover a potential augmentation of the existing $n$-conjunction.

considering only average performance. Once the new rule $\mathcal{A}_{U+1}$ is found, we add it to the rule set $R^{U+1} \leftarrow R^{(u)} \cup \{\mathcal{A}_{U+1}\}$.

The algorithm stops until no new rule can significantly improve the objective or after a predefined number of iterations. The algorithm outputs the final rule set along with their weights. After solving the master problem, we may choose to eliminate rules with weights below a specified threshold, ensuring that only the most impactful rules remain in the set, thereby improving the algorithm's efficiency and effectiveness. The pseudo-code and additional details are provided in Appendix A.

## 6 EXPERIMENT

To validate the effectiveness of our proposed method, this experimental section begins with evaluations of synthetic datasets with clearly defined predicates, ground truth consumer types, and decision rules. We then assess distribution learning and event prediction accuracy using two complex real-world datasets. The detailed ways we predefine the predicates for the real dataset is shown in appendix A. Our method is systematically compared against several baseline approaches to provide a comprehensive performance evaluation.

### 6.1 EXPERIMENTAL SETUP

**Baselines** We employ MNL and Mix-MNL (Jagabathula et al., 2020) as representatives of the classic models, while Vanilla NN, TasteNet (Han et al., 2020), DeepMNL (Wang et al., 2020; Sifringer et al., 2020), Mix-DeepMNL (Jagabathula et al., 2020), and RUMNet (Aouad & Désir, 2022) serve as neural network-based alternatives. A comprehensive description and comparison of models on utility value calculation, complexity, and interpretability is provided in Table 4 on Appendix B.

**Metric** We report the training and testing losses as negative log-likelihoods (Equation 3) and the top-1 accuracy for both datasets, where top-1 accuracy is the proportion of instances where the most selected product is predicted to have the highest choice probability. All results are reported as the average over 10 runs.

### 6.2 SYNTHETIC DATA

**Generation of Offersets and Choice Data** A total of 5,000 offer sets are generated, each comprising 20 products. Each product is characterized by 50 synthetic binary features sampled from a Bernoulli distribution, representing the grounded features of 50 product predicates. The complete set of predicates associated with products can be denoted as $\mathcal{P}^c = \{p_m^c\}_{m=1}^{50}$. Additionally, each offer set is associated with 15 binary features induced by offerset predicates, denoted as $\mathcal{P}^o = \{p_m^o\}_{m=1}^{20}$. These types of predicates are frequently encountered in practical scenarios and universal for all choice in offerset, such as search criteria in a query or the age of a patient requiring medication. The total sales volume $N_t$ for each offer set is fixed as 100 units. Three distinct preference types are predefined. Each type follows three decision rules, i.e. conjunctions of predicates. Only the first 10 product predicates and 2 offer set predicates are utilized within these preference types, adhering to the sparsity of effective predicates and demonstrating the model's selectivity when faced with redundant predicates. The choice result is computed as $100 \cdot mixed\ choice\ probability$.

**Rule Comparison Metric** To facilitate the comparison of rules, we use **Jaccard index** between the rules' predicate sets, defined as follows.

$$J(A, B) = \frac{|A \cap B|}{|A \cup B|} \tag{7}$$

$|\cdot|$ refers to the set size, $A$ and $B$ refer to the predicate sets of the two rules. For example, the Jaccard index of the following two rules $p_1 \wedge p_2 \wedge p_3$ and $p_4 \wedge p_2 \wedge p_3$ is 0.5, with $A \cap B = \{p_2, p_3\}$ $|A \cap B| = 2$ and $A \cup B = \{p_1, p_2, p_3, p_4\}$ $|A \cup B| = 4$. This metric directly shows the difference between the rule contents.

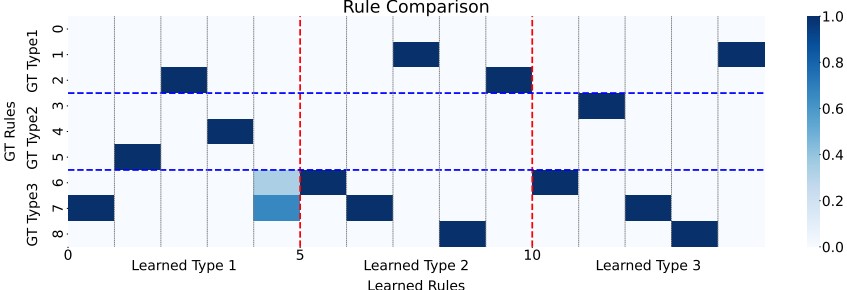

Figure 3: *Rule Comparison*. The figure shows the Jaccard index between the ground truth (GT) rules (Y-axis) and the rules learned by our risk seeking model (X-axis), denoted by the color intensity.

| Ground Truth Type | Ground Truth Rules | Learned Rules | Jaccard Index |
|---|---|---|---|
| | $p_1^c$ | / | 0 |
| Type 1 | $p_2^c \wedge p_3^c$ | $p_2^c \wedge p_3^c$ | 1 |
| | $p_4^c \wedge p_5^c \wedge p_6^c$ | $p_4^c \wedge p_5^c \wedge p_6^c$ | 1 |
| | $p_7^c$ | $p_7^c$ | 1 |
| Type 2 | $p_8^c \wedge p_9^c \wedge p_{10}^c$ | $p_8^c \wedge p_9^c \wedge p_{10}^c$ | 1 |
| | $p_{11}^c \wedge p_1^o$ | $p_{11}^c \wedge p_1^o$ | 1 |
| | $p_{12}^c$ | $p_{12}^c$ | 1 |
| Type 3 | $p_{13}^c \wedge p_{14}^c$ | $p_{13}^c \wedge p_{14}^c$ | 1 |
| | $p_{15}^c \wedge p_2^o$ | $p_{15}^c \wedge p_2^o$ | 1 |

Table 1: *Ground truth rules and closest learned rules on the synthetic dataset.*

**Results and Discussion** Figure 3 illustrates the comparison of learned rules with respect to the ground truth rules. Rule sets associated with the top-3 preference types, regarding their proportions, are considered as learned rules. A comprehensive comparative analysis between the ground truth rules and the identified rules is provided in Table 1. Remarkably, eight out of nine ground truth rules are perfectly learned. The results demonstrate that Logic-Logit effectively and selectively captures the majority of the rules without being encumbered by other irrelevant predicates included within the data. Fig4 provides a more intuitive result visualization.

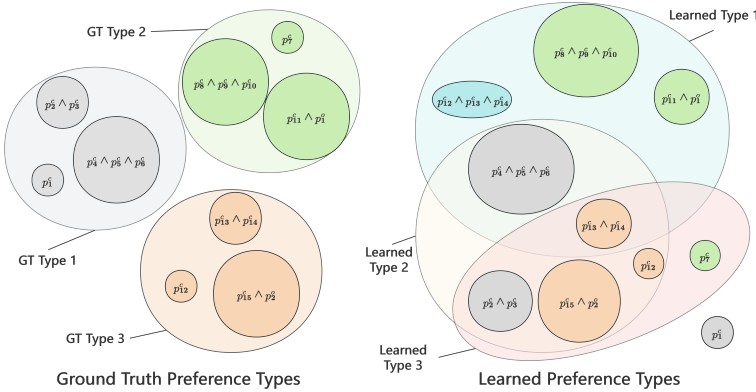

Figure 4: Rule Coverage

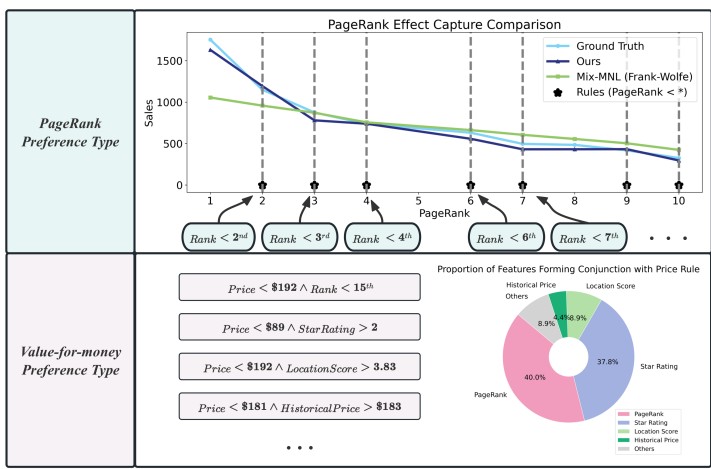

Figure 5: Preference Type Analysis of Expedia Hotel Dataset

## 6.3 EXPEDIA HOTEL DATASET EXPERIMENT

**Dataset Introduction** The Expedia Hotel Dataset[1] is released in 2013 on a Kaggle competition for improving the recommendation system of Expedia. The dataset is characterized by a structured format comprising both hotel search queries and corresponding search outcomes. Each search instance in the dataset consists primarily of two key components: the search criteria and the characteristics of the retrieved hotel results. The set of hotels retrieved through a search can be conceptualized as an 'offerset', wherein each hotel represents an option. The hotel that the customer ultimately books can be considered as the selected choice within this set. The detailed feature of each instance and experiment setting is shown in AppendixD.

**Results and Discussions** Figure 5 illustrates the two primary preference types identified in the Expedia Hotel Dataset: the *PageRank Preference Type* (85%) and the *Value-for-Money Preference Type* (15%). It is important to note that this does not imply that 85% of customers strictly follow PageRank when selecting hotels; rather, individuals typically exhibit characteristics of both types, with PageRank serving as a significant factor influencing choice behavior.

The ability of Mix-MNL and Logic-Logit to capture the PageRank effect is shown by comparing the aggregated choice probabilities of options at different ranking positions with actual sales data. Logic-Logit fits rules to model sales increases from lower to higher ranks. For example, the rule Rank $< 3^{rd}$ is derived from the substantial sales difference between hotels ranked $1^{st}$ and $2^{nd}$ versus those ranked $3^{rd}$ to $10^{th}$. In scenarios where sales remain consistent across ranks, such as between $4^{th}$ to $6^{th}$ and $7^{th}$ to $9^{th}$, no rules are detected. Overall, Logic-Logit segments the PageRank into distinct stages with defined rules and models the non-linear effect by assigning varying weights to these rules, whereas Mix-MNL primarily captures a linear effect.

Furthermore, the analysis of the Value-for-Money preference type indicates that customers associate price with multiple factors, including PageRank, star rating, location score, and historical price, through conjunction rules. Unlike neural network-based approaches that encode features into complex latent layers, Logic-Logit explicitly captures the correlations between features in an interpretable manner. A comprehensive performance comparison between Logic-Logit and baseline methods using the Expedia hotel dataset is included in Appendix C.

## 6.4 MIMIC-IV DATASET EXPERIMENTS

MIMIC-IV[2] is an electronic health record dataset of patients admitted to the intensive care unit (ICU) (Johnson et al., 2023). We considered patients diagnosed with sepsis (Saria, 2018), one of the major causes of mortality in ICU due to septic shock. Septic shocks are medical emergencies and early recognition and treatment would improve survival. We aim to uncover the treatment choice behavior of doctors.

---

[1]https://www.kaggle.com/datasets/vijeetnigam26/expedia-hotel
[2]https://mimic.mit.edu/

**Treatments and Vital Signs**   Suggested by (Komorowski et al., 2018), we extracted 14 treatments categorized as vasopressors, antibiotics, and auxiliary treatments associated with sepsis. We then select 16 vital signs that are highly related to sepsis. A detailed introduction to extraction and data processing can be found in Appendix E.

| Drug Name | Detected Rules |
|---|---|
| Norepinephrine | $PT > 12.6 \wedge ABPd \leq 59.0$ |
|  | $PTT > 31.6 \wedge age \leq 76.0 \wedge ABPd <= 54.0$ |
| Vasopressin | $PCO2(Arterial) > 41.0$ |
|  | $INR > 1.1 \wedge anchor_a ge \leq 66.0$ |
| Packed Red Blood Cells | $ABPd \leq 57.0 \wedge ABPm \leq 56.0$ |
|  | $PT > 12.6 \wedge INR > 1.4 \wedge RR \leq 32.0$ |
|  | $PlateletCount \leq 95.0 \wedge PTT > 31.6$ |

Table 2: *MIMIC Drug Preference Rules Examples.*

**Results and Discussion**   In the context of drug prediction tasks, the feature set of the data is predominantly centered on patient characteristics rather than the attributes of the drugs being considered. Consequently, classical linear models such as Multinomial Logit (MNL) and Mixed Multinomial Logit (Mix-MNL) are ill-equipped to address this task effectively. Table 3 shows NN-based models also perform badly due to the lack of product features. Furthermore, the medical domain necessitates decision-making that is interpretable and traceable. Our model is capable of extracting the underlying logical rules and their associated weights that govern physicians' medication selection processes.

Table 11 presents exemplar rules that support drug preference learning, consistent with established medical knowledge. Taking packed red blood cells as an illustrative example, the rule $ABPd \leq 57.0 \, \text{mmHg} \wedge ABPm \leq 56.0 \, \text{mmHg}$ indicates that the patient's blood pressure is below the normal threshold. Additionally, elevated levels of PTT, PT, and INR, along with a deficiency in platelet count, suggest that the patient is unable to achieve hemostasis. These rules collectively indicate that the patient is severely injured and experiencing significant blood loss, which provides interpretable evidence for the doctor's preference for packed red blood cells.

| Category | Method | Train loss | Test loss | Train accu (Top1) | Test accu (Top1) |
|---|---|---|---|---|---|
| DeepNN | NN | 2.5118(±0.137) | 2.5215(±0.033) | 0.1261(±0.043) | 0.0727(±0.031) |
|  | TasteNet | 2.4223(±0.000) | 2.4323(±0.000) | 0.1079(±0.008) | 0.0727(±0.064) |
|  | DeepMNL | 2.3696(±0.000) | 2.4331(±0.000) | 0.1102(±0.008) | 0.0818(±0.065) |
|  | mixDeepMNL | 2.3394(±0.000) | 2.4672(±0.000) | 0.1060(±0.008) | 0.0848(±0.065) |
|  | RUMNet | 2.3530(±0.000) | 2.4426(±0.000) | 0.1008(±0.002) | 0.1000(±0.064) |
| Ours* | Ours-NM-50 | 2.098(±0.0165) | 2.2047(±0.020) | 0.2961(±0.010) | 0.264(±0.029) |
|  | Ours-RS-50 | 2.0791(±0.011) | 2.1713(±0.025) | 0.3269(±0.003) | 0.26(±0.016) |

Table 3: *Treatment choice prediction on MIMIC-IV dataset.* We report Negative log likelihood (NLL) loss and Top-1 choice prediction accuracy of the training and testing set. NM refers to the normal mode of our model, while RS indicates the risk-seeking mode. We extract 50 rules for both types. Our Logit-Logit model performs much better than all the baselines.

## 7    CONCLUSION

In this study, we have proposed the Logic-Logit model, which incorporates logic rules into choice modeling as a pivotal strategy for enhancing both interpretability and accuracy. Our approach utilizes column generation for effective rule learning and leverages the Frank-Wolfe algorithm to facilitate the learning of mixture distributions for preference types. The empirical results obtained from real-world datasets demonstrate that our model significantly outperforms traditional neural network models in terms of both interpretability and predictive accuracy. This finding underscores the potential of integrating logical reasoning into choice models to produce more transparent and reliable decision-making tools.

ACKNOWLEDGMENT

Shuang Li's research was in part supported by the NSFC under grant No. 62206236, Shenzhen Stability Science Program 2023, Shenzhen Science and Technology Program ZDSYS20230626091302006, Longgang District Key Laboratory of Intelligent Digital Economy Security, and SRIBD Innovation Fund SIF20240010.

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

APPENDIX OVERVIEW

In the following, we will provide supplementary materials to better illustrate our methods and experiments.

- Section A presents more details of our model and implementation.
- Section B introduces the baseline methods we considered in our paper.
- Section C provides more additional experiment results about synthetic datasets.
- Section D provides more additional experiment results about Expedia Hotel datasets.
- Section E provides more details about MIMIC-IV dataset experiments.

# A  IMPLEMENTATION DETAILS

## A.1  PREDEFINING THE PREDICATES

The specification of predicates can be quite flexible. Any method generate binary label from the original choice feature can be used as predicates, such as traditional classification model. Here we introduce the most direct way to achieve this, which is also used in the two real dataset experiments.

**Discrete Features**  Discrete features inherently possess finite conditions due to the limited set of possible values that the feature can assume, such as $if\ number\ of\ rooms = 3$ in a house. If house containing one to ten rooms appear in the dataset, then there will naturally be ten predicates indicating how many rooms are in a house.

**Continuous Features**  The possible value of a continuous feature is infinite, thus we cannot use the equivalence condition like discrete feature. We have continued with the approach of Wei et al. (2019) to get predicates by thresholding the feature with the percentiles within the dataset, such as $if\ house\ price\ ranks\ in\ top\ 20\%$ and $if\ house\ area\ fail\ to\ rank\ in\ top\ 60\%$. The number of these percentile thresholds can be determined as a hyperparameter. Dense thresholds bring about accuracy for the rule content, but also lead to larger searching space and difficulty for the rule searching step. This methodology is based on the distribution of the feature, and can eliminate the influence of outlier features and feature scales, which may cause trouble in linear models.

## A.2  SUBPROBLEM SOLVING: RULE SEARCHING WITH COLUMN GENERATION

The pseudo-code of the rule searching step is shown below:

---

**Algorithm 1** Solving Frank-Wolfe Step using Column Generation

---
1: **Input:** $max\ conjunction\ length\ K$
2: **Initialize:** $k = 1, ruleset\ R = \{\}, rule\ weight\ \boldsymbol{w} = []$
3: **while** $stopping\ condition\ 1\ not\ met$ **do**
4:     **while** $k \leq K$ **do**                                          ▷ BFS
5:         $q \leftarrow 0$
6:         **while** $stopping\ condition\ 2\ not\ met$ **do**
7:             $\Delta R = Generate\ Candidate\ Rules(R, k, q)$
8:             Compute $\nabla\phi(\boldsymbol{w}_{new})$ for each new rule in $\Delta R$
9:             $\Delta R^* \leftarrow$ Select Top-N potential Rules from $\Delta R$ with $\nabla\phi(\boldsymbol{w}_{new})$ information
10:            $R \leftarrow R \cup \Delta R^*$
11:            $\boldsymbol{w} \leftarrow \arg\min_{\boldsymbol{w}} \phi\,(\boldsymbol{w} \mid R)$
12:            $q \leftarrow q + 1$
13:        **end while**
14:        $k \leftarrow k + 1$
15:    **end while**
16: **end while**
17: **return** $ruleset\ R, rule\ weight\ \boldsymbol{w}$

---

The algorithm 1 shows the entire road map of Solving Frank-Wolfe Step using Column Generation, including the rule searching and potential evaluation with gradient. The stopping condition 1 controls the number of total preference types to be found. It can be set as a predefined number or depends on the convergence of NLL loss. The stopping condition 2 controls how many rules with searching length will be searched in a preference type. It can also be set as a predefined number or

depends on the convergence of Frank-Wolfe step loss (as show in Eq. 5). Suppose the current rule set is $R^{(U)}$, the detailed formulation for computation of the $\nabla\phi(\boldsymbol{w}_{new})$ is shown below:

$$\nabla\phi(\boldsymbol{w}_{new}) = -\frac{1}{N}\sum_{t=1}^{T}\sum_{j\in S_t}\left(\frac{N_{jt}}{g_{jt}^{(k-1)}}\right)\cdot\left(\mathbb{I}_{new}(\boldsymbol{x}_{jt}) - \sum_{\ell\in S_t}\frac{\exp\left(z_{\ell t}\right)}{\sum_{\ell'\in S_t}\exp\left(z_{\ell' t}\right)}\cdot\mathbb{I}_{new}(\boldsymbol{x}_{\ell t})\right)$$

where $z_{\ell t} = \sum_{u=1}^{U} w_u\cdot\left(\bigwedge_{p_m\in\mathcal{A}_u} p_m\left(\boldsymbol{x}_{\ell t}\right)\right)$, and $g_{jt}^{(k-1)}$ refers to the choice probability of product $j$ in offer set $S_t$ mixed by the previous found $k-1$ preference types. With weights of rules that have already been in rule set fixed, we initialize the candidate rules weights $\boldsymbol{w}_{new}$ to be zeros, and calculate the gradient of loss upon this weights. In other words, we use the gradient to determine if changing the weight of the candidate rule from zero can bring about improvement in Frank-Wolfe step solving. The absolute value of the gradient directly implies the potential of a candidate rule. Therefore we can select top-n most potential rules and put them in $R$.

---

**Algorithm 2** Generate Candidate Rules

---

1: **Input:** $R, k, q$
2: **Initialize:** $\Delta R = \{\}$
3: **if** $q = 0$ **then**
4:     **for** $rule \in R$ **do**
5:         **if** $rule\ conjunction\ length = k - 1$ **then**
6:             Extend rule conjunction with each predicate respectively         $\triangleright$ DFS
7:             Put all extended rules in $\Delta R$
8:         **end if**
9:     **end for**
10: **else**
11:     Randomly Sample $k$ predicates and form conjunction
12:     Repeat line 11 for multiple times and put generated rules in $\Delta R$
13: **end if**
14: **return** $\Delta R$

---

From Algorithm 1 and Algorithm 2, we can find out that the our model searches rules following the order of rule conjunction length, i.e. BFS. Each time the model finishes length-k rule searching then start length-k+1 rule searching. Moreover, to improve the rule searching efficiency, we also introduce DFS when the searching change from length-k to length-k+1. For example, when we start to search for length-2 rule we first go through all the length-1 rule in the current rule set and form conjunctions with other predicates to get length-2 rules induced by length-1 rules in current rule set. This comes from the simple intuition that choices satisfy condition $A \wedge B$ must satisfy condition $A$ and condition $B$ simultaneously. Although single predicate $A$ doesn't depict the $A \wedge B$ perfectly, increasing the weight of predicate $A$ can still increase the utility of choices satisfy $A \wedge B$ and bring about improvement.

### A.3   OPTION FEATURE AND OFFER SET FEATURE

Our model distinguishes between option-specific features and offerset-specific features, treating them differently. The offerset features are exclusively considered within the context of conjunction rule searching. Any rule composed solely of offerset features is skipped during the rule searching process due to the inherent structure of the logit. Irrespective of the weights assigned to these rules, they are inevitably discarded during the softmax process since offerset features are common across all options within the same offerset. Consequently, rules involving offerset features are only functional when they are conjunct with option-specific features. The rules pertaining to option features elucidate the attractiveness of individual options, whereas the rules associated with offerset features delineate the scenarios (or offersets) in which various option-specific rules should be applied. This nuanced approach enables Logic-Logit to successfully incorporate offerset features into choice modeling, thereby enhancing its explanatory power and predictive accuracy.

### A.4   RULE SET EXPANSION, RE-WEIGHTING, AND PRUNING

The rule space to be explored is extensive, necessitating careful control of the rule set size for efficient subproblem resolution. An overly large rule set can lead to time-consuming optimization and compromise interpretability.

**Rule Set Expansion**  During a single subproblem iteration, a substantial number of rules are identified, many of which exhibit promise. However, only the top-k most promising rules are incorporated into the rule set. This strategy curbs the expansion rate of the rule set, maintaining its sparsity and interpretability.

**Rule Set Pruning**  Our algorithm employs two phases of pruning. The first occurs during subproblem resolution, while the second is executed post-resolution.

During subproblem resolution, to ensure the problem remains tractable, we impose a maximum rule set size, typically set generously. When the size exceeds this threshold, rules with low weights are discarded. Post-discard, the remaining rules undergo re-weighting.

Following subproblem resolution, to uphold interpretability and sparsity, an ultimate maximum size limit is enforced, significantly smaller than during the subproblem phase. Similarly, rules are pruned, and the retained rules are re-weighted.

**Re-weighting**  For each preference type, the number of rules must be manageable to preserve interpretability. Selecting the most significant rules is critical during the re-weighting process. We apply a relatively high $l1$ penalty when optimizing rule weights to ensure sparsity. Notably, this penalty is not applied post the second pruning phase, as the rule set size is already constrained by the ultimate maximum limit.

## A.5 Computing Infrastructure

Our experiments are applicable to both CPU and GPU. For this paper, we use GPU NVIDIA GeForce RTX 3090.

## A.6 Optimizer

We choose the Adam optimizer to do stochastic gradient descent in both rule weights learning and preference type proportion update.

## B Baselines

- **Classic Choice Models**
    - MNL (McFadden, 1981) Multinomial logit model
    - Mix-MNL (Jagabathula et al., 2020) MNL with feature weights belonging to finite discrete support learned with Frank-Wolfe algorithm framework.

- **Deep Learning-Driven Choice Models**
    - Vanilla NN is a simple and standard feed-forward neural network that takes a concatenated vector of all product and customer attributes as input, and outputs the utility of each choice alternative. These utility values are then passed through a softmax layer to calculate the choice probabilities for each alternative.
    - TasteNet (Han et al., 2020) uses deep learning to model individual tastes/preferences directly in a choice modeling context.
    - DeepMNL (Wang et al., 2020) combines the traditional MNL model with deep learning architectures to capture more complex choice behavior.
    - MixDeepMNL (Jagabathula et al., 2020) is a mixture of DeepMNL models that allow for heterogeneity in preferences across individuals.
    - RUMNet (Aouad & Désir, 2022) is a deep learning-based model that combines random utility theory with neural networks. It provides an efficient approximation of random utility maximization (RUM) discrete choice models.

**Hyperparameters setting of Deep learning driven choice models**  We apply similar hyperparameters for all neural network-based models followed the setting in RUMNet (Aouad & Désir, 2022). We select the standard ADAM optimizer with a learning rate of 0.001 and batch sizes of 32. We set label smoothing as 0.0001, a norm-based regularization method on the neural network's outputs. We tune the neural network structure and report the best one in all tables. In particular, for each model, we vary the parameters $(l, w) \in (3, 10), (5, 20), (10, 30)$, where $l$ denotes the depth of the network and $w$ denotes its width.

| Model | Utility Value | # Parameters | Grounds for Interpretability |
|---|---|---|---|
| **Logic-Logit (Ours)** | Rule-Based | $\mathcal{O}(Nn)$ | Rules Content & Weights |
| MNL | Linear | $\mathcal{O}(d)$ | Feature Weights |
| Mix-MNL | | $\mathcal{O}(Nd)$ | |
| Vanilla NN | NN-Based | $\mathcal{O}(dd_v + Ld_v^2)$ | / |
| TasteNet | | | |
| Deep-MNL | | | |
| Mix-DeepMNL | | | |
| RUMNet | | | |

Table 4: *Model Comparison*, In the # parameter column, $N$ represents the consumer type number, $n$ denotes the maximum rule number, and $d$ indicates the dimensionality of the product feature vector. For NN-based models, $d_v$ are dimensions of latent features and $L$ is the number of layers. As the inherent complexity of NN-based models, their interpretability is often considered negligible.

## C  DETAILED EXPERIMENTS RESULTS

In this section, we present the detailed experimental results by comparing our model with several baseline approaches. In ours, NM refers to the normal mode of our model, while RS indicates the risk-seeking mode. The number at the end denotes the count of rules for each preference type.

| Category | Method | Train loss | Test loss | Train accu (Top1) | Test accu (Top1) |
|---|---|---|---|---|---|
| Classic | MNL | 2.7503($\pm$0.003) | 2.7751($\pm$0.005) | 0.16($\pm$0.002) | 0.1312($\pm$0.007) |
| | Mix-MNL (Frank-Wolfe) | 2.7472($\pm$0.003) | 2.7679($\pm$0.004) | 0.1631($\pm$0.002) | 0.1382($\pm$0.011) |
| DeepNN | NN | 2.9671($\pm$0.016) | 3.0075($\pm$0.007) | 0.0739($\pm$0.008) | 0.0553($\pm$0.014) |
| | TasteNet | 2.7178($\pm$0.022) | 2.7511($\pm$0.054) | 0.1759($\pm$0.012) | 0.1620($\pm$0.027) |
| | DeepMNL | 2.4603($\pm$0.024) | 2.6001($\pm$0.040) | 0.2564($\pm$0.004) | 0.2099($\pm$0.006) |
| | mixDeepMNL | 2.4538($\pm$0.021) | 2.5798($\pm$0.057) | 0.2404($\pm$0.004) | 0.2253($\pm$0.019) |
| | RUMNet | 2.4444($\pm$0.050) | 2.5749($\pm$0.071) | 0.2442($\pm$0.012) | 0.2186($\pm$0.048) |
| Ours* | Ours-NM-5 | 2.6032($\pm$0.017) | 2.6397($\pm$0.023) | 0.2151($\pm$0.010) | 0.2048($\pm$0.011) |
| | Ours-NM-20 | 2.5242($\pm$0.0069) | 2.5871($\pm$0.0098) | 0.2204($\pm$0.0034) | 0.1876($\pm$0.007) |
| | Ours-RS-5 | 2.5926($\pm$0.008) | 2.6305($\pm$0.014) | 0.2208($\pm$0.005) | 0.2058($\pm$0.011) |
| | Ours-RS-20 | 2.5215($\pm$0.0047) | 2.5836($\pm$0.0041) | 0.2231($\pm$0.0019) | 0.1908($\pm$0.0064) |

Table 5: Choice prediction results on the synthetic dataset. We report Negative log likelihood (NLL) loss and Top-1 choice prediction accuracy of the training and testing set. NM refers to the normal mode of our model, while RS indicates the risk-seeking mode. We extract 5 / 20 rules for both types.

The synthetic data is generated with single sales, implying that within every offerset, only one option is selected. The Logic-Logit model demonstrates comparable performance in terms of both Negative Log-Likelihood (NLL) Loss and Top1 accuracy. Notably, the risk-seeking mode model yields improved outcomes compared to the normal mode, as it aids the model in fitting to uncovered samples, maintaining an equilibrium between exploration and exploitation.

In both the Expedia Hotel Dataset and the MIMIC Dataset, our model demonstrates superior performance over neural network-based models and classic models in terms of loss and accuracy.

Within the hotel dataset, it is evident that increasing the maximum rule number enhances the expressiveness of the Logic-Logit model and improves its fit to the training data, although this may lead to over-fitting. Notably, the risk-seeking heuristic proves ineffective in this context due to the dominance of preference-type sales and the PageRank effect. Enhancing rule diversity does not yield significant performance benefits.

The MIMIC dataset predominantly comprises patient features, specifically offerset-specific features shared among all drug options. Consequently, we primarily focused on models capable of learning both drug and patient features. Classic linear models and Vanilla Neural Networks are not suitable to this scenario.

| Category | Method | Train loss | Test loss | Train accu (Top1) | Test accu (Top1) |
|---|---|---|---|---|---|
| Classic | MNL | 3.1482($\pm$0.029) | 3.1423($\pm$0.032) | 0.177($\pm$0.020) | 0.1595($\pm$0.005) |
| | Mix-MNL (Frank-Wolfe) | 2.9654($\pm$0.023) | 2.9576($\pm$0.025) | 0.1807($\pm$0.008) | 0.1657($\pm$0.006) |
| DeepNN | NN | 3.0991($\pm$0.028) | 3.1669($\pm$0.023) | 0.2168($\pm$0.001) | 0.2120($\pm$0.011) |
| | TasteNet | 3.0909($\pm$0.017) | 3.0863($\pm$0.015) | 0.1503($\pm$0.007) | 0.1431($\pm$0.013) |
| | DeepMNL | 3.0494($\pm$0.011) | 3.0696($\pm$0.022) | 0.1566($\pm$0.002) | 0.1586($\pm$0.007) |
| | mixDeepMNL | 3.0283($\pm$0.020) | 3.0755($\pm$0.025) | 0.1620($\pm$0.010) | 0.1507($\pm$0.010) |
| | RUMNet | 3.0182($\pm$0.002) | 3.0736($\pm$0.020) | 0.1631($\pm$0.004) | 0.1489($\pm$0.008) |
| Ours* | Ours-NM-30 | 2.9505($\pm$0.019) | 2.9754($\pm$0.013) | 0.1901($\pm$0.004) | 0.1698($\pm$0.009) |
| | Ours-RS-30 | 2.9473($\pm$0.017) | 2.9675($\pm$0.016) | 0.1909($\pm$0.002) | 0.1738($\pm$0.005) |

Table 6: Choice prediction on the Expedia Hotel dataset. We report Negative log likelihood (NLL) loss and Top-1 choice prediction accuracy of the training and testing set. Our Logit-Logit model performs best. NM refers to the normal mode of our model, while RS indicates the risk-seeking mode. We extract 30 rules for both types.

# D   ADDITIONAL EXPEDIA HOTEL EXPERIMENTS

**Feature Introduction**   The Expedia Hotel Dataset[3] is a publicly available dataset released on Kaggle in 2013. It records customers' search-and-booking interaction with Expedia platform. The features listed in table 7 are used in the experiment.

| Feature Type | Feature Name | Variable Type | Description |
|---|---|---|---|
| Hotel Feature | PageRank | Discrete | The recommendation position of the hotel |
| | Star Rating | Continuous | The historical star rating |
| | Location Score | Continuous | The desirability of a hotel's location |
| | Historical Price | Continuous | Mean price of the hotel over the last period |
| | Branded | Binary | If the hotel is part of a major hotel chain |
| | Promotion | Binary | If the hotel had a sale price promotion |
| | Price | Continuous | Displayed price of the hotel |
| | Booking | Binary | If customer book the hotel |
| Search Criterion | Booking Window | Discrete | Number of days in the future from search date |
| | Length of Stay | Discrete | Number of nights stay |
| | Adults Count | Discrete | The number of adults |
| | Children Count | Discrete | The number of children |
| | Room Count | Discrete | The number of rooms |
| | Saturday Night | Binary | If the stay includes a Saturday night |
| | Random | Binary | If the displayed sort is random |

Table 7: Descrption of feature description on the Expedia Hotel dataset.

**Experiment Setting**   To prepare our dataset, we divided it into training and testing subsets using a 3:1 ratio based on search timestamps. We selected offersets with 28 to 32 search results, whose results is not randomly displayed, i.e. sorted by Expedia Recommendation System. Searches with no bookings, hotels priced over \$1000, or booking windows longer than one year were excluded. Finally, we randomly sampled 10,000 instances for the training set and 1,000 instances for the test set.

**Model Setting**   The hyper parameters of our model is set as below. This setting is shared for Expedia Hotel experiments in both Appendix D and Appendix C.
- Overall Convergence Criterion: The NLL loss decreases less than 0.001 after a new preference type is found and proportion is updated.
- The ultimate output rule number for each preference type is set to be 30 for Table 6, 30 and 100 for Table 8.
- The maximum rule number of the rule set (Rule Prune Threshold) $R$ during column generation is set to be 100.

---

[3]https://www.kaggle.com/datasets/vijeetnigam26/expedia-hotel

- The candidate rule set size generated during the column generation iteration is set to be 100 * searching rule length.
- The number of percentile thresholds we put on the continuous features to get predicates is set to be 30, i.e. a threshold per 3.3%.
- We search rules up to 3 conjunctions. For each length-1 to length-3 conjunctions we search for 10, 50, 100 iterations.

**Additional Large-Scale Expedia Hotel Dataset Experiments**   Moreover, in order to show our model's scalability on bigger dataset, we also evaluate our model with 50,000 training instances and 5000 testing instances, containing 15-32 search results

| Category | Method | Train loss | Test loss | Train accu (Top1) | Test accu (Top1) |
|---|---|---|---|---|---|
| Classic | MNL | 2.9525(±0.134) | 2.9578(±0.131) | 0.1633(±0.042) | 0.1607(±0.042) |
| | Mix-MNL (Frank-Wolfe) | 2.8267(±0.029) | 2.8341(±0.028) | 0.1883(±0.014) | 0.1875(±0.013) |
| DeepNN | NN | 3.1317(±0.015) | 3.2699(±0.011) | 0.1138(±0.003) | 0.0863(±0.003) |
| | TasteNet | 2.8278(±0.032) | 2.8384(±0.025) | 0.1877(±0.007) | 0.1885(±0.009) |
| | DeepMNL | 2.7722(±0.022) | 2.7833(±0.030) | 0.1999(±0.006) | 0.2020(±0.007) |
| | mixDeepMNL | 2.7453(±0.025) | 2.7717(±0.015) | 0.2036(±0.004) | 0.2035(±0.008) |
| | RUMNet | 2.7244(±0.009) | 2.7532(±0.016) | 0.2105(±0.003) | 0.2072(±0.002) |
| Ours* | Ours-NM-30 | 2.8258(±0.001) | 2.8310(±0.002) | 0.2015(±0.000) | 0.1958(±0.001) |
| | Ours-NM-100 | 2.8014(±0.008) | 2.8166(±0.005) | 0.2058(±0.002) | 0.1997(±0.001) |
| | Ours-RS-30 | 2.8281(±0.003) | 2.8328(±0.003) | 0.2013(±0.000) | 0.1952(±0.001) |
| | Ours-RS-100 | 2.7983(±0.002) | 2.8165(±0.003) | 0.2067(±0.000) | 0.1992(±0.002) |

Table 8: Choice prediction on larger scale Expedia Hotel dataset. We report Negative log likelihood (NLL) loss and Top-1 choice prediction accuracy of the training and testing set. We extract 50/100 rules for both types.

In large-scale datasets, our model performs comparably to NN-based models, excelling in-sample accuracy over the majority of NN-based models. Logic-Logit, with significantly fewer parameters compared to neural network models, simultaneously demonstrates superior performance and rule-based interpretability.

## E   ADDITIONAL MIMIC-IV EXPERIMENTS

MIMIC-IV[4] is a publicly available database sourced from the electronic health record of the Beth Israel Deaconess Medical Center (Johnson et al., 2023). Available information includes patient measurements, orders, diagnoses, procedures, treatments, and deidentified free-text clinical notes. Sepsis is a leading cause of mortality in the ICU, particularly when it progresses to septic shock. Septic shocks are critical medical emergencies, and timely recognition and treatment are crucial for improving survival rates. In the real-world healthcare data experiments on the MIMIC-IV dataset, we aim to understand how doctors choose the treatments to diagnose.

**Patients**   We select 3000 patients that satisfied the following criteria from the dataset: (1) The patients are diagnosed with sepsis (Saria, 2018). (2) Patients, if diagnosed with sepsis, the timestamps of any clinical testing and timestamps of medication administration and corresponding dosage were not missing.

**Vital signs**   Referred to (Komorowski et al., 2018), we collected 16 vital signs that are highly related to sepsis from chart events recorded in the ICU. Table 10 shows these vitals with names and descriptions.

**Treatment**   Suggested by (Komorowski et al., 2018), we extracted 21 treatments associated with sepsis which are consistent with expert consensus. Based on the distinct clinical characteristics of these treatments, they can be categorized into the following three groups, which are shown in Table 9. Vasopressor therapy is a fundamental treatment of septic-shock-induced hypotension as it aims at correcting the vascular tone depression and then improving organ perfusion pressure; Antibiotics

---

[4] https://mimic.mit.edu/

also should be given within a few hours of the diagnosis of sepsis; Some auxiliary treatments such as packed red blood cells and invasive ventilation are also necessary in ICU.

**Data Preprocessing**    For each patient, we extract the treatments belonging to the above three categories, as well as the corresponding treatment start and end times. Then, we select the patient's vital features during the treatment. We only record the first vital value. During this period, some vital values will be lost, so we only keep the most complete records as the processed dataset. Another problem is label imbalance, so we only keep 14 treatment types with enough records, and if the number of some treatment types is too large, we will delete some records. Finally, we randomly selected 3000 records as the training set and 300 records as the test set.

**Experiment Settings**    The hyper parameters of our model is set as below:
- Overall Convergence Criterion: The NLL loss decreases less than 0.005 after a new preference type is found and proportion is updated.
- The ultimate output rule number for each preference type is set to be 50.
- The maximum rule number of the rule set (Rule Prune Threshold) $R$ during column generation is set to be 100.
- The candidate rule set size generated during the column generation iteration is set to be 100 * searching rule length.
- The number of percentile thresholds we put on the continuous features to get predicates is set to be 20, i.e. a threshold per 5%.
- We search rules up to 5 conjunctions. For each length-2 to length-5 conjunctions we search for 50, 100, 150, 200 iterations.

**Sensitive analysis**    We conduct sensitive analysis for three significant hyper-parameters, **ultimate rule number** for each preference type, the **maximum conjunction length** and **rule prune threshold** we search up to in the rule searching with MIMIC dataset.

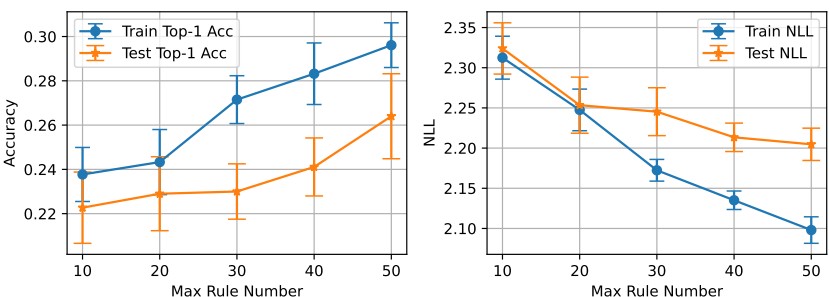

Figure 6: *Max Rule Number*

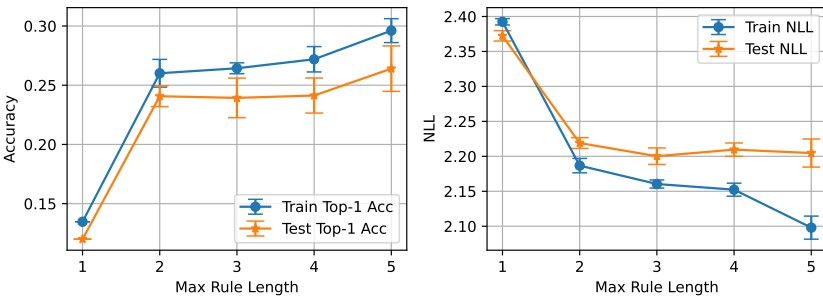

Figure 7: *Max Rule Length*

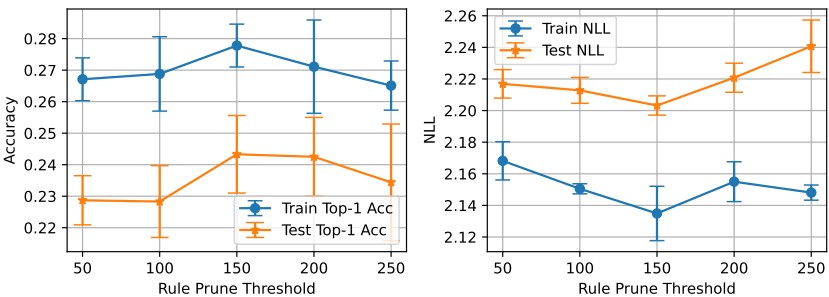

Figure 8: *Rule Prune Threshold*

Fig 6 and Fig 7 shows that the model performance improves significantly with the max rule number and max rule length. Both of these two hyperparameters increase the complexity of each preference type's rule set. For algorithmic point of view, increasing the maximum complexity of the ruleset will enlarger feasible space $P$ for the functional Frank-Wolfe step solving and bring about improvement for the whole model. Meanwhile increasing the rule set complexity will also bring about additional computational cost, a trade-off should be made when tuning these hyperparameters. Setting the ultimate rule number to be 50, Fig8 shows the relationship between the rule prune threshold and model performance. Setting a larger rule prune threshold than the maximum rule number allows for a wider tolerance range for newly searched rules, making it easier for promising rules to be learned. However, if this value is set too high, excessive deletion of redundant rules during the final pruning phase can result in significant changes in the ruleset, leading to a decline in model performance. Therefore, it is generally recommended to set the rule prune threshold to 1.5-2 times the maximum rule number.

| Category | Treatment |
|---|---|
| **Vasoconstrictor** | Epinephrine |
| | Phenylephrine |
| | Norepinephrine |
| | Dobutamine |
| | Dopamine |
| | Vasopressin |
| | Angiotensin II (Giapreza) |
| **Antibiotic** | Vancomycin |
| | Caspofungin |
| | Cefepime |
| | Ceftriaxone |
| | Gentamicin |
| | Micafungin |
| | Tobramycin |
| | Piperacillin/Tazobactam |
| **Auxiliary Treatment** | Furosemide (Lasix) |
| | Heparin Sodium |
| | Invasive Ventilation |
| | Packed Red Blood Cells |
| | IV Immune Globulin (IVIG) |
| | Acetaminophen-IV |

Table 9: Description of the treatment extracted from MIMIC-IV dataset.

| Item id | Abbreviation | Description | Normal Range |
|---------|--------------|-------------|--------------|
| 220546 | WBC | White blood count | $4,000 - 10,000 cells/mm^3$ |
| 225690 | Total Bilirubin | Total Bilirubin | 0.1 - 1.2 mg/dL |
| 220235 | PCO2 (Arterial) | Partial pressure of carbon dioxide (Arterial) | 35 - 45 mmHg |
| 226063 | PO2 (Venous) | Partial pressure of oxygen (Venous) | 30 - 40 mmHg |
| 229761 | Creatinine (whole blood) | Creatinine (whole blood) | 0.5 - 1.2 mg/dL |
| 226253 | SpO2 Desat Limit | Oxygen saturation (SpO2) | 95% - 100% |
| 220210 | RR | Respiratory rate | 12 - 20 breaths per minute |
| 220052 | ABPm | Arterial Blood Pressure mean | 70 - 100 mmHg |
| 220051 | ABPd | Arterial Blood Pressure diastolic | 60 - 80 mmHg |
| 223762 | Temperature C | Temperature Celsius | 36.1 - 37.2 |
| 220045 | HR | Heart rate | 60 - 100 beats per minute |
| 220050 | ABPs | Arterial Blood Pressure systolic | 90 - 120 mmHg |
| 227465 | PT | Prothrombin time | 11 - 13.5 seconds |
| 227467 | INR | International normalized ratio | 0.8 - 1.2 |
| 227466 | PTT | Partial thromboplastin time | 25 - 35 seconds |
| 227457 | Platelet Count | Platelet Count | 150 - 450 $\times 10^9$ platelets/L |

Table 10: Description of the vital signs extracted from MIMIC-IV dataset. Item id is the ID extracted from MIMIC-IV tables.

| Drug Name | Detected Rules |
|-----------|----------------|
| Norepinephrine | $PT > 12.6 \land ABPd \leq 59.0$ 
 $PTT > 31.6 \land age \leq 76.0 \land ABPd <= 54.0$ |
| Vasopressin | $PCO2(Arterial) > 41.0$ 
 $INR > 1.1 \land anchor_age \leq 66.0$ |
| Epinephrine | $RR > 20.0 \land WBC <= 6.8 \land INR <= 3.4$ 
 $ABPs \leq 77.0$ |
| Furosemide (Lasix) | $ABPs > 93.0 \land WBC \leq 14.55$ 
 $ABPs > 107.0$ |
| Dobutamine | $INR <= 1.4 \land PCO2(Arterial) \leq 36.0$ 
 $PCO2(Arterial) <= 51.0 \land ABPs <= 89.0$ |
| Dopamine | $PTT > 65$ 
 $PlateletCount <= 287.0 \land PTT > 48.8$ |
| Packed Red Blood Cells | $ABPd \leq 57.0 \land ABPm \leq 56.0$ 
 $PT > 12.6 \land INR > 1.4 \land RR \leq 32.0$ 
 $PlateletCount \leq 95.0 \land PTT > 31.6$ |
| Antibiotic | $age \leq 66.0 \land RR \leq 30.0 \land WBC \leq 20.6$ 
 $PCO2(Arterial) <= 42.0$ |

Table 11: *MIMIC Drug Preference Rules Examples*

