# OpenReview forum: "Logic-Logit: A Logic-Based Approach to Choice Modeling"
_ICLR.cc/2025/Conference — ICLR 2025 Poster_

### Official Review · Reviewer_brV8 · 2024-10-27

**Soundness:** 2
**Presentation:** 3
**Contribution:** 2
**Rating:** 5
**Confidence:** 3

**Summary:**

The paper introduces the Logic-Logit, an interpretable, rule-based choice model, along with a learning pipeline built on column generation techniques and the Frank-Wolfe algorithm. The authors also provide numerical results demonstrating the model's interpretability and predictive accuracy.

**Strengths:**

The paper is generally well-written and presents a new explainable choice model with a learning framework. Numerical results indicate strong interpretability and predictive performance, tested on both synthetic and real-world datasets.

**Weaknesses:**

[a]There is no discussion regarding the model's capacity. It is unclear whether this model falls under the mixed-logit model or if it is equivalent to a mixed-logit model. If so, why not use the approach in [1] to learn a mixed-logit, given its stronger analytical properties, such as provable convergence guarantees? (Also, the mixed-logit's identified customer segments and the parameters in each segment could also be viewed as the interpretation of the choice?)

[b]I'm confused about if the learning requires prior knowledge of $M$ and $\{p_m\}$. For instance, Line 177 mentions, “These mappings, from real-valued features to predicates, are predefined and fixed,” and the learning problem (3) does not appear to learn $\{p_m\}$. Yet, in the experiments, the algorithm identifies these rules. This raises questions about the rule search strategies outlined in lines 308-323, and a pseudo-code would help clarify this process.

[c]The literature review lacks a discussion of consider-then-choose models. Since the proposed model follows this approach, it would be beneficial to compare it with other models in this category, such as [2] and [3].

Minor comments:

[a]Given that interpretability is a key advantage of the proposed model, it would be helpful to compare it with other interpretable choice models, such as [4].

[b]Including the variance/std in the experimental results would provide additional insight.

[c]Some potential typos: line 192 should be “item $s$ ” instead of “item $j$”, and also “offer set $S$” instead of “offer set $S_t$”.


References

[1]Hu, Yiqun, David Simchi-Levi, and Zhenzhen Yan. "Learning mixed multinomial logits with provable guarantees." Advances in Neural Information Processing Systems 35 (2022): 9447-9459.

[2]Liu, Qing, and Neeraj Arora. "Efficient choice designs for a consider-then-choose model." Marketing Science 30.2 (2011): 321-338.

[3]Akchen, Yi-Chun, and Dmitry Mitrofanov. "Consider or Choose? The Role and Power of Consideration Sets." arXiv e-prints (2023): arXiv-2302.

[4]Tomlinson, Kiran, and Austin R. Benson. "Learning interpretable feature context effects in discrete choice." Proceedings of the 27th ACM SIGKDD conference on knowledge discovery & data mining. 2021.

**Questions:**

See weaknesses [a] [b].

---

> ### Author Response · Authors · 2024-12-01
> **Response to Reviewer brV8**
>
> We appreciate that reviewer brV8 has a positive impression of the model’s interpretability and predictive performance. To address the questions about the details of our method, we provide point-wise responses as follows.
>
> **It is unclear whether this model falls under the mixed-logit model or if it is equivalent to a mixed-logit model. If so, why not use the approach in [1] to learn a mixed-logit, given its stronger analytical properties, such as provable convergence guarantees?**
>
> Thanks for your valuable questions and advice. Our model is equivalent to a special case of a mixed-logit model with exponentially large numbers of binary features. Each feature is induced by a conjunction of predicates, i.e. a rule. Because of the large feature size, it’s hard for us to directly learn a taste vector on it. Thus we use column generation to do rule searching which can avoid the curse of dimension, keep the sparsity of rules, and maintain interpretability.
>
> We appreciate the reviewer’s insightful reference to the improved Frank-Wolfe algorithm for learning mixed MNL models, as proposed in [1].
> The Q construction algorithm in [1] restricted the searching space for the choice probability vector in a $\sigma$ ball. It is powerful in preventing the original Frank-Wolfe algorithm from learning boundary types (Jagabathula et al. [2020]) and helps specify an individual's decision mode with better convergence guarantees.
> Even though the algorithm is powerful, it has two key restrictions:
>
> 1) The algorithm highly relies on **panel data**, the information of individual decision-makers at different time. The construction of Q needs multiple choice data for one individual decision-maker to calculate the similarity between individuals. However, this is not a general setup in the practice dataset. Like the Expedia hotel dataset, most of the users (identified by the user ID) only have a few choices in the dataset.
>
> 2) Moreover, what the algorithm actually does is find multiple subsets of decision makers with similar decision modes, i.e. construct a small population base for further finding Q. The problem lies in following steps that the algorithm needs to learn the taste vector for every subset of decision makers’ choosing results, i.e. MNL parameter estimation, and map the taste vector back to a choice probability vector. Then the vector serves as candidate support in support finding steps. In our model setting, we need to do a column generation for N times, assuming the number of decision-maker subsets is N. This will intensively increase the computational complexity for each Frank-Wolfe step and significantly harm the model’s learning speed.
>
> Thus the approach in [1] is not suitable for our model. Our model focuses on revealing the possibility of combining the interpretable rule learning mechanism with the choice model. We want to show that such a combination works under common choice data settings without additional restrictions. Thus we choose to use the original Frank-Wolfe proposed by (Jagabathula et al. [2020]), instead of [1]. But introducing [1]’s Q construction method in our model will be a very promising direction for further work on this topic to get individual decision rules specified and provable guarantee, even though this may only work well when the predicates set is relatively small.
>
> **Also, the mixed-logit's identified customer segments and the parameters in each segment could also be viewed as the interpretation of the choice?**
>
> Thanks for the question. We will briefly compare the interpretability of the mixed-logit models and our model, which can be summarized as three points.
>
> The mixed-logit model computes the utility value of a choice linearly, and the coefficients of different features can be interpreted. However, when the feature dimension is large, the dense taste vector may not be easily interpreted. In contrast, our model preserves the sparsity of predicates and maintains interpretability through rules.
>
> Secondly, our model also captures nonlinear, interpretable effects between features—for instance, in the house motivation example, decision-makers can consider multiple predicates within a single conjunction.
>
> Furthermore, the linear utility function in mixed-logit models cannot incorporate customer-specific information. It’s because customer features are identical across all choices and are thus ignored during the softmax computation. Thanks to the nonlinear utility calculation in our model, conjunctions between customer predicates and choice predicates are effectively captured. This leads to interpretable logic statements such as, “When the decision-maker is (some situation), they will prefer choices that satisfy *** conditions.” Therefore, our model offers significantly stronger interpretability compared to the normal mixed-logit model.

---

> ### Author Response · Authors · 2024-12-01
>
> **I'm confused about if the learning requires prior knowledge of M and Pm. For instance, Line 177 mentions, “These mappings, from real-valued features to predicates, are predefined and fixed,” and the learning problem (3) does not appear to learn Pm. Yet, in the experiments, the algorithm identifies these rules.**
>
> Thanks for the valuable question pointing out a potentially confusing point in our paper. The predefined predicates can be quite flexible. Any labeling from the choice feature to binary {0, 1} can be a predicate, such as classification models or output of label propagation models in GNN. The way to get predicates is not the focus point of this paper.
> In our paper, we use an intuitive and interpretable way to define the predicates on real datasets, which is also added in Appendix A: Predefining the Predicates:
>
> Discrete features such as categorical features inherently possess finite conditions due to the limited set of possible values that the feature can assume, such as if the number of rooms = 3 in a house. If a house containing one to ten rooms appears in the dataset, then there will naturally be ten predicates indicating how many rooms are in a house.
>
> We binarize continuous features with specific thresholds. In order to distribute the threshold uniformly in the data, we utilize the percentile of the feature. If we need 5 thresholds for feature i, we first aggregate the feature i from every choice in every offersets (all the products appear in the dataset)  as a set, then 20% 40% 60% 80% percentiles of this set can be easily calculated. Therefore the predicate’s mapping condition can be interpreted as “if the choice ranks in the top 20% regarding feature i”.
>
> This way of defining predicates can capture the sudden non-linear shift of utility value brought about by the continuous feature, such as the page rank effect we show in the Expedia Dataset Experiments. Details about predefining the predicates is also included in Appendix A after revision.
>
> **This raises questions about the rule search strategies outlined in lines 308-323, and a pseudo-code would help clarify this process.**
>
> For the Frank-Wolfe Step Solving with Column Generation,  we added the pseudo-code of the algorithm for Rule Searching with Column Generation in Appendix A.
>
> **There is no discussion regarding the model's capacity**
>
> Our rule-based methodology employs a process conceptually akin to basis discovery, where the model sequentially uncovers fundamental patterns, or “basis,” that serve to explain the data.  Specifically, our approach identifies rules incrementally by discovering simpler, interpretable components (or "basis") from the data, which are then combined to generate more complex, high-level rules. This sequential discovery allows the model to avoid overwhelming complexity by starting from large predicate set and gradually scaling up. The model's capacity is therefore flexible, as it can progressively capture more complex patterns by refining and combining these basis elements.

---

> ### Author Response · Authors · 2024-12-01
>
> **The literature review lacks a discussion of consider-then-choose models. Since the proposed model follows this approach, it would be beneficial to compare it with other models in this category, such as [2] and [3].**
>
> Thanks for the advice, helping us build a bridge between the consider-then-choose model and our model.  We add a part for discussion about consider-and-choice models in the related work section 2.1 of the revised manuscript. Moreover, we would like to add more discussions here.
> Consider-then-choice (CTC) models describe a two-stage decision-making process:
>
> 1) **Consideration Stage:** Consumers use non-compensatory rules (e.g., conjunctive or disjunctive screening) to form a subset of alternatives based on criteria like price, brand, or ratings.
>
> 2) **Choice Stage:** A compensatory model evaluates the filtered alternatives to select the final choice.
>
> Several studies have emphasized the importance of constructing the consideration set in multi-stage decision models. Liu et al. [1] developed a statistical method for two-stage consider-then-choose models, focusing on conjunctive screening rules, where a product is considered only if all its features meet specific criteria. They highlight the value of integrating prior knowledge into the design process. Akchen et al. [2] approached consideration sets as nonparametric models defined by distributions over sets, capturing stochastic consumer behavior. They simplified the choice stage by assuming uniform random selection within the consideration set.
>
> Other works address heterogeneity and inconsistencies. For instance, [3] introduced a latent consideration set choice model that accommodates randomness and variability in screening and choice behavior. Traets et al. [4] accounted for individual thresholds in screening, modeling variability in how people filter alternatives using conjunctive rules. Similarly, [5] included taste heterogeneity to enhance e-commerce assortment personalization, improving both user satisfaction and revenue. Finally, Aouad et al. [6] proposed a dynamic programming framework to optimize assortments efficiently by breaking the problem into smaller subproblems.
> Our proposed model aims to provide a structured framework for capturing human choice preference patterns. Unlike the studies mentioned earlier, our approach leverages column generation and employs OR-of-ANDs logic rules to address the complexities of understanding human decision-making and extracting underlying rules.
>
> In future work, we plan to include more comparative baseline experiments. However, in the current experimental setup, our model is not fully comparable to the two-stage models discussed, as we assume individuals make choices directly from all available alternatives rather than selecting from a carefully calculated subset.
>
> [1] Liu, Qing, and Neeraj Arora. "Efficient choice designs for a consider-then-choose model." Marketing Science 30.2 (2011): 321-338.
> [2] Akchen, Yi-Chun, and Dmitry Mitrofanov. "Consider or Choose? The Role and Power of Consideration Sets." arXiv e-prints (2023): arXiv-2302.
> [3] Assele, Samson Yaekob, Michel Meulders, and Martina Vandebroek. "The value of consideration data in a discrete choice experiment." Journal of choice modelling 45 (2022): 100374.
> [4] Traets, Frits, Michel Meulders, and Martina Vandebroek. "Modelling consideration heterogeneity in a two-stage conjunctive model." Journal of Mathematical Psychology 109 (2022): 102687.
> [5] Li, Maggie Manqi, et al. "Integrating empirical estimation and assortment personalization for e-commerce: A consider-then-choose model." Xiang and Huang, Yan and Shi, Cong, Integrating Empirical Estimation and Assortment Personalization for E-Commerce: A Consider-Then-Choose Model (September 10, 2018) (2018).
> [6] Aouad, Ali, Vivek Farias, and Retsef Levi. "Assortment optimization under consider-then-choose choice models." Management Science 67.6 (2021): 3368-3386.

---

### Official Review · Reviewer_7vMh · 2024-10-29

**Soundness:** 3
**Presentation:** 1
**Contribution:** 2
**Rating:** 5
**Confidence:** 3

**Summary:**

This paper addresses the development of choice models by constructing a rule-based approach using logical statements that dissect conjunctions. The proposed method employs a dual-optimization process: an outer optimization for determining preference weights, and an inner optimization to identify new rules. The model iteratively refines the rule set by incorporating each newly discovered rule. Assuming convex optimization, the authors apply the Frank-Wolfe algorithm to update preference weights across all rule types. Experiments on a synthetic dataset and the Expedia Hotel Dataset demonstrate that the proposed method consistently outperforms all baseline models.

**Strengths:**

- The authors tackle a crucial question in designing interpretable and explainable models.
- They present an interpretable algorithm tailored for choice modeling.
- The method is intuitive and demonstrates strong performance.
- The proposed approach outperforms baseline methods across 2 datasets.

**Weaknesses:**

- The method does not appear scalable to a large number of features (predicates). What is the computational complexity of this approach?

- The experimental details, including hyperparameters (e.g., exact convergence condition used) for the proposed algorithm, are missing and were not found in the Appendix. Significant methodological details are lacking.

- Although logic rules enhance interpretability, they may not be applicable for designing all types of features.

**Questions:**

- Item $j$ and $S_t$ are introduced but not used in Equation (1), which creates some confusion.
- If $p_m$ returns 1 when a condition is met and $x_s$ is a single item, does that mean the resulting conjunctions reduce to just $x_s$?
- The implementation details of BFS and DFS in the algorithm are unclear, as no specifics or pseudocode are provided. Pseudocode of the algorithm would be helpful with all conditions.
- The notation $\mathcal{X}$ is used but not defined.
- Some figures (1,2) are included but not referenced within the text.
- Could you clarify the rationale behind the rule distance metric? How would the distance between two unrelated features be interpreted?
- Why do the neural network (NN) models lack product features?
- Why aren’t there results for NN-based models on the synthetic dataset?
- The paper does not provide details on how the baseline methods were trained. What are their hyperparameters and how were they trained?
- Other discrete choice models, such as graph-based approaches [1], mixed logit models, and network formation models [2], are not included for comparison, which might help comparing tradeoffs between methods.

[1] Tomlinson, Kiran, and Austin R. Benson. "Graph-based methods for discrete choice." Network Science 12.1 (2024): 21-40.

[2] Gupta, Harsh, and Mason A. Porter. "Mixed logit models and network formation." Journal of Complex Networks 10.6 (2022): cnac045.

---

> ### Author Response · Authors · 2024-12-01
> **Response to Reviewer 7vMh**
>
> We appreciate Reviewer 7vMh’s valuable feedback. We are delighted that the reviewer recognize the strengths of our approach in terms of interpretability and performance. To address the questions and concerns about our method, we provide point-wise responses as follows.
>
> **The experimental details, including hyperparameters (e.g., exact convergence condition used) for the proposed algorithm, are missing and were not found in the Appendix. Significant methodological details are lacking.**
>
> We would like to express our gratitude to Reviewer 7vMh for pointing out the issue of lacking crucial methodological details. In response, we have now incorporated the essential information regarding our model settings in Appendix D for the Expedia Dataset and Appendix F for the MIMIC dataset.
>
> **Although logic rules enhance interpretability, they may not be applicable for designing all types of features.**
>
> We appreciate the insights provided by Reviewer 7vMh regarding the applicability of logic rules in designing various types of features. In Appendix A, we elaborate on our methodology for creating predicates from both discrete and continuous features, which we believe addresses this concern effectively.
>
>
> 1 ) For **discrete features**, such as categorical variables, the inherent finite conditions stem from the limited set of possible values the feature can assume. For instance, in a house dataset, if the number of rooms can range from one to ten, there would naturally be ten predicates representing the number of rooms in a house (e.g., "if the number of rooms = 3").
>
>
> 2 ) In the case of **continuous features**, we binarize them using specific thresholds. To ensure a uniform distribution of these thresholds across the data, we employ the percentile of the feature. For example, if we require five thresholds for feature i, we aggregate feature i from every choice in every offer set (comprising all products in the dataset) to form a set. We then calculate the 20%, 40%, 60%, and 80% percentiles of this set. Consequently, the predicate's mapping condition can be interpreted as "if the choice ranks in the top 20% (for example) regarding feature i."
>
> Furthermore, the generation of predicates can be made more flexible through more complex methods, such as classification models, enhancing the model's applicability to sophisticated data forms.
>
> **Item $j$ and $S_t$ are introduced but not used in Equation (1), which creates some confusion.**
>
> **The notation $X$ is used but not defined.**
>
> **Some figures (1,2) are included but not referenced within the text.**
>
> Thank you very much for your thorough review and for identifying the typos in our manuscript. Your meticulous attention to detail is greatly appreciated. The typos are revised in the new version. After revision, we found the $x_{s}$ in the numerator on the right side of Equation (1) should be changed to $x_{j}$. We sincerely apologize for this oversight.
>
> **If $p_m$ returns 1 when a condition is met and $x_s$ is a single item, does that mean the resulting conjunctions reduce to just $x_s$?**
>
> We appreciate the insightful comments provided by the reviewer. However, we believe this question requires further clarification to ensure accurate addressing. The conjunction of predicates maps to binary values (0 or 1), rather than a feature vector. Below is an example that demonstrates how we calculate an item’s utility value using rules, which may aid in understanding.
>
> Our algorithm directly utilizes rules to determine utility values, offering a flexible approach to specifying the consideration set. Only items that meet certain condition rules receive a bonus to their utility value. For instance, consider an item with the feature vector $[x^1, x^2] = [10, 20]$. The rules are defined as follows:
> - Rule A: $x^1 > 5$
> - Rule B: $x^2 < 10$
> - Rule C: $x^1 < 15 \wedge x^2 > 15$
>
> The weights assigned to these rules are 1, 2, and 3, respectively. Upon evaluation, the item satisfies Rule A and Rule C, but not Rule B. Thus, the utility value for the item is calculated as follows:
> \[ \text{Utility Value} = 1 \times 1 + 2 \times 0 + 3 \times 1 = 4 \]
>
> **The implementation details of BFS and DFS in the algorithm are unclear, as no specifics or pseudocode are provided. Pseudocode of the algorithm would be helpful with all conditions.**
>
> We have included pseudocode in Appendix A detailing the solution for the Frank-Wolfe step with column generation and its subsequent formulation. We appreciate this valuable advice.

---

> ### Author Response · Authors · 2024-12-01
>
> **Could you clarify the rationale behind the rule distance metric? How would the distance between two unrelated features be interpreted?**
>
> We appreciate the reviewer for pointing out the potential issues with our rule comparison metric. Following revisions, we have adopted a new synthetic experimental setup that directly synthesizes predicates instead of using continuous or binary features. (Additionally, Appendix A now includes a discussion on how to define predicates from real data.)
>
> This new approach allows for a more straightforward comparison of rules. Recall that the rules we search using column generation take the form of conjunctions with predicates. By treating two rules as two sets of predicates, we can compare the ground truth rules with the learned ones using the Jaccard Index. The formula for the Jaccard Index, J(A, B), is defined as $ \frac{|A \cap B|}{|A \cup B|} $. For instance, the Jaccard index of the following two rules $ p_1 \wedge p_2 \wedge p_3 $ and $ p_4 \wedge p_2 \wedge p_3 $ is 0.5, where $ A \cap B = \{ p_2, p_3 \} $ and thus $ |A \cap B| = 2 $, while $ A \cup B = \{p_1, p_2, p_3, p_4\} $ and thus $ |A \cup B| = 4 $. This metric clearly illustrates the content differences between the rules.
>
> **Why do the neural network (NN) models lack product features?**
>
> In the MIMIC drug prediction dataset, the features of choice mainly focus on depicting the situation of the patient, i.e. the offer set features. Under this circumstance, a good and flexible choice model should find out the relationship between the drug choice and the patient's physical signs. However,  we find out that the NN models perform badly on offer-set-feature-rich datasets like MIMIC, but normally on choice-feature-rich datasets like Expedia Hotel. This point also shows our model’s flexibility and robustness when facing different datasets.
>
> **Why aren’t there results for NN-based models on the synthetic dataset?**
>
> Most of the NN-based models can only support input offer sets where only one option is chosen. This creates difficulty for directly using the synthetic data described in the main text of the paper to evaluate the NN-based models. To address this challenge, we change the setting of the chosen result for comparison in the Appendix. We first calculate the ground truth choice probability for options in every offset and randomly sample one of the options to be chosen according to the choice probability of each option. Then with the new setting, we can compare our model and NN-based models. We have included the results for NN-based models in the revision.
>
> **The paper does not provide details on how the baseline methods were trained. What are their hyperparameters and how were they trained?**
>
> We appreciate the reviewer's valuable advice. Additional details regarding this aspect are provided in Appendix B of the revised manuscript.
>
> **Other discrete choice models, such as graph-based approaches [1], mixed logit models, and network formation models [2], are not included for comparison, which might help comparing tradeoffs between methods.**
>
> We would like to express our gratitude to the reviewer for suggesting the inclusion of additional discrete choice models for comparison, which could indeed offer valuable insights into the tradeoffs between different methods.
>
>
> 1. **Graph-Based Approaches:** While graph-based approaches are powerful, they are not directly applicable to our problem setting. These models, such as those requiring prior graph knowledge, do not align with the intrinsic nature of our problem. However, we acknowledge that these models could provide potential future directions for enhancing our algorithm by incorporating more sophisticated structural insights.
>
>
> 2. **Mixed Logit Models:** We have already included the mixed logit model in our baseline classic algorithms, utilizing the Frank-Wolfe algorithm.
>
> We appreciate the reviewer's suggestions and believe that while some models may not be directly comparable due to differing problem settings, they provide valuable insights that could guide future work.

---

> ### Comment · Reviewer_7vMh · 2024-12-01
> **Thank you for your response!**
>
> I would like to thank the authors for answering my questions and greatly updating the paper.
>
> Thank you for updating the paper with additional information on hyperparameters. In Appendix D, Table 8, although accuracy and loss are not directly dependent on each other, I am wondering why with increasing loss (train and test), the accuracy drops.
>
> Additionally, my concern is still not addressed about designing logic rules for complex features. Regardless of being discrete or continuous, if some predicaments are challenging in raw (or numerical) classification (e.g., text), the additional external model needs to be trained for better classification, which adds additional complexity to the method. Furthermore, for continuous features, it is even harder to decide how many predicaments the user needs to create (as it can reach to infinity). But then again, it leads to the complexity issue that I raised in my review. Unfortunately, I do not see a simple solution to that yet.

---

> > ### Author Response · Authors · 2024-12-02
> >
> > Thank you for your insightful questions! We provide responses as follows:
> >
> > **Although accuracy and loss are not directly dependent on each other, I am wondering why with increasing loss (train and test), the accuracy drops.**
> >
> > We appreciate your observation on the relationship between the negative log likelihood (NLL) loss and top-1 accuracy. The loss function utilized in this paper is indeed the NLL loss, where an increase in loss signifies a decline in prediction performance. Consequently, it is expected that a rise in loss will correspond with a decrease in accuracy. The top-1 accuracy is the proportion of instances where the most selected product is predicted to have the highest choice probability. This metric alone does not fully describe the accuracy of learning the choice probability distribution. Therefore, it is acceptable to observe minor anomalies in the relationship between top-1 accuracy and NLL loss.
> >
> > **Concerns about designing logic rules for complex features**
> >
> > We appreciate the reviewer’s concerns about designing logic rules for complex features.
> >
> > Let’s take the textual data as an example. We can use existing models, such as NLP models, to obtain the sentiment tendency of the text (happy or sad), whether it exhibits aggressiveness, and which topic it belongs to. These can transform complex data types that are difficult to process directly into predicates that can be accepted by our model. The training of such models is not based on the logic-logit and can be trained independently, thus does not affect the complexity of our choice model. On the contrary, the introduction of these models brings the possibility for our model to handle more data types, such as text data.
> >
> > Regarding continuous features, we acknowledge the difficulties in determining the number of thresholds and the associated computational complexity inherent in the method described in our article. However, it is crucial to emphasize that our objective is to harmonize interpretability with practical applicability within the choice model framework.
> >
> > The fundamental motivation for utilizing logic rules is to ensure transparency in decision-making processes. For instance, in product recommendation systems, logic rules enable both users and system designers to comprehend the rationale behind specific product suggestions. An example might be, “if the product price falls within the top 20% and the user rating exceeds 4,” thereby enhancing user trust and engagement with the model. Interpretability becomes particularly vital when model decisions influence user choices, as it fosters trust and improves the adoption of the system.
> >
> > Interpretability by itself does not necessitate an infinite number of thresholds. Our empirical results further demonstrate that a limited number of thresholds are sufficient to achieve performance on real datasets that is comparable to that of NN-based models.

---

> > > ### Comment · Reviewer_7vMh · 2024-12-02
> > > **Thank you for your response!**
> > >
> > > I would like to thank the authors for their reply!
> > >
> > > Since the authors has responded to my other concerns, I am raising my score.
> > >
> > > However, the weaknesses of the method still remain (W1 and W3).

---

> > > > ### Author Response · Authors · 2024-12-04
> > > >
> > > > Thank you for your efforts during the discussion phase. We are glad that we have addressed your concerns. Regarding Weaknesses 1 and 3, we acknowledge potential limitations. Due to the iterative nature of column generation, the computational cost is high during the training phase, and we indeed have to balance interpretability and computational complexity. The issue of complex features stems from the question how to get predicates from raw data, which is important but not central to this paper. We believe it will be better addressed in future work. We are grateful for your relevant questions and look forward to potential further discussions with you in the future.

---

### Official Review · Reviewer_59ua · 2024-11-08

**Soundness:** 3
**Presentation:** 3
**Contribution:** 2
**Rating:** 6
**Confidence:** 2

**Summary:**

This paper introduces an approach to modeling human choice using OR-of-ANDs logic rules. The authors illustrate how any preference can be transformed into a Boolean logic formula, with rules mapped to Boolean space and interconnected through AND and OR operators. This formula is then incorporated into a mixed logit choice model, enabling preference learning. However, this approach results in an infinite-dimensional optimization problem, a major computational challenge. To address this, the authors apply the functional conditional gradient method (Frank-Wolfe) to reduce the optimization’s complexity. Additionally, due to the exponential size of the search space ($2^M - 1$, where $M$ is the number of rules), they use a column generation technique that incrementally expands the search space by adding new rules in each step. Empirical experiments on both synthetic and real-world datasets (from commercial and healthcare domains) highlight the effectiveness and versatility of this approach.

**Strengths:**

1. The paper is well-organized and motivated. Its focus on interpretable human preferences is valuable for understanding human decision-making, particularly in high-stakes areas like healthcare and autonomous driving, where trust and transparency are essential.
2. The approach to modeling human choice could also inspire advancements in fields like automated reasoning (the task authors have verified), where a clear understanding of decision-making processes is crucial.

**Weaknesses:**

1. While innovative, the current solution appears complex, particularly due to the combined use of the functional conditional gradient method and column generation. This complexity may limit its applicability or make implementation challenging for practitioners. A more streamlined or efficient approach could enhance the method’s usability across a wider range of real-world applications.
2. The search space of combinatorial rules is increased exponentially number of rules $M$, which needs to be further reduced to accelerate the learning process.

**Questions:**

1. Could you estimate the computational cost of the proposed methods, particularly for real-world datasets?
2. Are there specific conditions or data characteristics under which this model is computationally more efficient or challenging?

---

> ### Author Response · Authors · 2024-11-29
> **Response to Reviewer 59ua**
>
> We appreciate Reviewer 59ua’s positive feedback and the insight into how our approach could advance automated reasoning and decision-making. To address the questions and concerns about our method, we provide point-wise responses as follows.
>
>
>
> **While innovative, the current solution appears complex, particularly due to the combined use of the functional conditional gradient method and column generation. This complexity may limit its applicability or make implementation challenging for practitioners. A more streamlined or efficient approach could enhance the method’s usability across a wider range of real-world applications.**
> We greatly appreciate the reviewer's thoughtful consideration of the potential complexity our approach may present, particularly concerning the integration of the functional conditional gradient method and column generation.
>
>
> 1) While these methodologies do introduce a level of computational intensity during the training phase, we wish to underscore that **interpretability is our foremost design priority**, even if it means compromising on training efficiency. Our model is structured to ensure that the decisions it makes are transparent and understandable.
>
>
> 2) It is crucial to highlight that the computational demands are largely **confined to the training period**. Post-training, the inference process exhibits remarkable efficiency, facilitating implementation in practical, real-world applications. This balance allows for a robust model that can be efficiently used without sacrificing the interpretability that is essential for effective decision-making.
>
>
> By steadfastly maintaining interpretability within our model, our objective is to equip decision-makers with clear and reliable insights. We firmly believe that these benefits far surpass the additional complexities encountered during training.
>
> **The search space of combinatorial rules is increased exponentially number of rules M, which needs to be further reduced to accelerate the learning process.**
> We sincerely appreciate the reviewer's insightful consideration of the potential challenge arising from the exponentially large rule searching space.
>
>
> 1 ) **Regarding the number M**: The combinatorial rule searching space is determined by the number of predefined predicates M. In this paper, we employ a straightforward but somewhat cumbersome method to generate predicates (equality and thresholding), as outlined in Appendix A. Nonetheless, the generation of predicates is inherently flexible; any model with a (0, 1) binary labeling output can serve as a predicate, such as a classical classification model. By incorporating more efficient models to define interpretable predicates, we can achieve a more concise and simplified predicate set, thereby reducing M.
>
>
> 2 ) **Regarding the length of rule conjunctions**: To uphold interpretability, our approach generally does not necessitate very long conjunctions of rules, which inherently limits the searching space. For example, in the Expedia Hotel Dataset, we confine our search to rules with a length of up to 3, and in the MIMIC dataset, we search up to length-5 rules. Despite these constraints, our model exhibits robust performance, indicating that restricting the rule length to a short range suffices for effective performance.
>
> Through the integration of efficient predicate generation models and the imposition of rule length restrictions, we can notably reduce the rule searching space and expedite the learning process. This strategy not only enhances the computational efficiency but also ensures the model's interpretability and practical applicability.
>
> **Are there specific conditions or data characteristics under which this model is computationally more efficient or challenging?**
>
> When the choice features are complex and the predicates set is large, the rule searching process may become more challenging. This is primarily because the combinatorial space grows significantly, leading to increased computational demands during the training phase.
>
> However, this challenge can be ameliorated with better predicate generation processes. By employing more sophisticated techniques to generate predicates, we can potentially reduce the size of the predicates set and simplify the rule searching process, thereby enhancing computational efficiency. For instance, leveraging domain-specific knowledge or machine learning models to generate more concise and relevant predicates could streamline the algorithm’s operation.
>
> In summary, while complex features and large predicates sets pose some computational challenges, advancements in predicate generation methods offer promising avenues to mitigate these issues and improve the overall efficiency of our model.

---

> ### Author Response · Authors · 2024-12-02
>
> **Could you estimate the computational cost of the proposed methods, particularly for real-world datasets?**
>
> Regarding computational cost, the training phase is the most demanding due to the iterative nature of column generation. Although column generation lacks a formal convergence rate, it converges in finite time, meaning it will eventually stop after a limited number of iterations. In practice, we can set stopping criteria based on factors like a maximum number of iterations or a threshold for improvement, allowing us to control the trade-off between solution quality and computational time.
>
> A comparison of algorithm run times on large-scale Expedia dataset is shown below:
> | Model      | Running time (s) |
> |----------------|---------------|
> | Ours Logic-Logit | 1208        |
> | DeepNN          | 69           |
> | TasteNet          | 88          |
> | DeepMNL          | 279           |
> | mixDeepMNL          | 975           |
> | RUMNet          | 9465           |

---

### Official Review · Reviewer_Xxs5 · 2024-11-09

**Soundness:** 3
**Presentation:** 3
**Contribution:** 3
**Rating:** 6
**Confidence:** 3

**Summary:**

This paper presents Logic-Logit, a rule-based interpretable choice model that utilizes logical rules to predict human choices in contexts like healthcare and commercial domains. The authors aim to address limitations in interpretability associated with existing neural network-based models by proposing a model that represents choices through OR-of-ANDs logic rules. These rules enable compact and interpretable representation of human decision-making. The paper introduces an optimization framework using the Frank-Wolfe algorithm combined with column generation to efficiently extract rules, showcasing empirical success in interpretability and accuracy across synthetic, commercial (Expedia Hotel), and healthcare (MIMIC-IV) datasets.

**Strengths:**

- The model’s combination of interpretable rule-based choice modeling with optimization algorithms is innovative. The approach’s focus on interpretable, structured rule extraction addresses a significant gap in choice modeling literature, especially relevant for high-stakes domains.
- The experimental setup is comprehensive, covering synthetic and real-world datasets. Benchmarks with traditional models and neural networks underscore the model’s effectiveness in balancing accuracy and interpretability. The rule extraction and optimization process is detailed and thoughtfully developed, providing clarity on algorithmic decisions.
- The paper is generally well-organized. The presentation of the OR-of-ANDs rule structure, Frank-Wolfe algorithm, and column generation steps is clear, supporting reproducibility. The inclusion of rule explanations on real-world datasets aids in understanding practical implications.

**Weaknesses:**

- While the column generation and rule pruning strategies manage computational demands, further discussion on the model’s scalability with significantly larger datasets would enhance the paper. For instance, scalability tests on larger commercial datasets would demonstrate practical feasibility in data-intensive domains.
- The approach involves selecting parameters like the number of rules, rule lengths, and pruning thresholds. More empirical insights into the sensitivity of these parameters on model performance, especially in healthcare contexts, could strengthen robustness claims.
- While the model shows good accuracy on average, there is less discussion about edge cases, where rule-based logic might oversimplify complex decision boundaries. Addressing potential limitations in handling such cases would provide a more balanced view of the model’s capabilities.

**Questions:**

- Can the authors provide more insight into how the model scales with an increase in the number of features or the size of the dataset? Would any adjustments be necessary in the Frank-Wolfe and column generation steps?
- How does the model handle cases where decision criteria overlap significantly between customer types? For instance, if preferences are highly correlated between types, does the model tend to overfit to certain rules or ignore relevant variation?
- Would incorporating neural components in conjunction with logic-based rules (e.g., neural embeddings for complex feature spaces) enhance performance without compromising interpretability? This hybrid approach could be of interest if straightforward rule-based methods struggle with nuanced distinctions in large feature sets.
- How can this method be applied for RLHF, can authors provide experimental results to demonstrate this? i.e., Finetuing an LLM for a Diffusion models.

---

> ### Author Response · Authors · 2024-11-29
> **Response to Reviewer Xxs5**
>
> We appreciate reviewer Xxs5’s affirmation of the innovativeness of our proposed model, the validity of our experiments, and the quality of our writing. To address the questions and concerns about our method, we provide point-wise responses as follows.
>
> **While the column generation and rule pruning strategies manage computational demands, further discussion on the model’s scalability with significantly larger datasets would enhance the paper. For instance, scalability tests on larger commercial datasets would demonstrate practical feasibility in data-intensive domains.**
>
> We thank the reviewer for their valuable advice on empirically testing our model’s scalability on a larger dataset. To address this point, we have conducted an additional experiment using **50,000 offer set instances sampled from the Expedia dataset**. This new experiment is five times larger than our previous ones, with each offer set containing an average of 27 products. The results of Table 8  in the revised manuscript are compared with our baseline models, clearly demonstrating our model’s scalability when confronted with a bigger dataset. These findings have been included in the revised manuscript in Appendix D.
>
> **The approach involves selecting parameters like, the number of rules, rule lengths, and pruning thresholds. More empirical insights into the sensitivity of these parameters on model performance, especially in healthcare contexts, could strengthen robustness claims.**
>
> An additional sensitive analysis of the model performance influenced by hyperparameters, the maximum number of rules, maximum rule lengths, and pruning thresholds is conducted with the MIMIC dataset. The results are shown in Appendix E.
>
> **Can the authors provide more insight into how the model scales with an increase in the number of features or the size of the dataset? Would any adjustments be necessary in the Frank-Wolfe and column generation steps?**
>
> We appreciate the reviewer’s concerns about the scalability. Our model’s scale with the number of features (predicates) and dataset size by enlarging:
>
> **1 ) Overall number of Preference Types we find**
> Most intuitively, the number of outer iterations we need for the model to converge will become larger, i.e. find more preference types.
>
> **2 ) The maximum number of ultimate rules contained by each preference type**
>  This helps when the number of predicates is relatively large and the decision process is complex, such as choice and decision in healthcare tasks. In our experiments, we set the number of rules for each preference type to be 50, larger than Expedia.
>
> **3 ) The max rule length of the rules we search during column generation**
> Recall that the max rule length restricts the number of conjunctions in rule searching. Enlarging it will bring about more information to each rule, which will help to enhance the model’s expressiveness.
>
> **4 ) The size of the candidate rule set we generate in every column generation iteration**
> This mainly helps the computational speed when the number of predicates is large. Recall that the potential calculation is just calculating the gradient, which can be sped up with GPU.
>
> **How does the model handle cases where decision criteria overlap significantly between customer types? For instance, if preferences are highly correlated between types, does the model tend to overfit to certain rules or ignore relevant variation?**
>
> This is a reason why we introduce the risk-seeking strategy in our algorithm, which forces the model to learn samples that haven't been covered well with previous learned preference types. Sometimes the learned preference type in risk-seeking only takes a small proportion compared with other preference types. However, this is still meaningful. The rules obtained in risk-seeking tell us that there exist other special rules that influence people’s decisions,  but may be not commonly seen in the population. This will give us interpretable insights for possible relevant variation.

---

> ### Author Response · Authors · 2024-11-29
>
> **Would incorporating neural components in conjunction with logic-based rules (e.g., neural embeddings for complex feature spaces) enhance performance without compromising interpretability? This hybrid approach could be of interest if straightforward rule-based methods struggle with nuanced distinctions in large feature sets.**
>
> At a low level, neural networks can function as powerful perceptrons, extracting intricate patterns and features from the data. To preserve interpretability, it is crucial to ensure that the low-level neural components are providing some labeled data, allowing them to perform tasks like concept extraction with clarity and justification.
>
> At a high level, the integration of rule-based systems can provide logical reasoning capabilities, ensuring that the decisions made by the model are transparent and understandable. This hierarchical approach—where low-level neural networks handle feature extraction and high-level rules guide decision-making—offers a balance between the flexibility of neural networks and the interpretability of rule-based systems [1] [2].
>
> This extension, while beyond the scope of this paper, presents a compelling direction for future research. By combining the strengths of both neural and rule-based methodologies, we can develop choice models that not only perform well but also remain interpretable, addressing the challenges posed by complex and large feature sets. This will be a promising direction for future work in interpretable choice models.
>
> **How can this method be applied for RLHF, can authors provide experimental results to demonstrate this? i.e., Fine Tuning an LLM for a Diffusion model.**
>
> The application of Logic-Logit in RLHF or LLM is not the focus point of this paper. However, a very recent work by OpenAI [3], reveals the possibility and potential of integrating rules in LLM and RLHF. A combination of rule-based human preference learning and LLM or RLHF may be a promising direction for future work.
>
> [1] Glanois, Claire, et al. "Neuro-symbolic hierarchical rule induction." International Conference on Machine Learning. PMLR, 2022.
>
> [2] Kusters, Remy, et al. "Differentiable rule induction with learned relational features." arXiv preprint arXiv:2201.06515 (2022).
>
> [3] Mu, Tong, et al. "Rule based rewards for language model safety." arXiv preprint arXiv:2411.01111 (2024)

---

### Author Response · Authors · 2024-11-29
**General Response**

Dear Esteemed Reviewers,

We sincerely thank you for dedicating your time and effort to reviewing our paper. Your thoughtful feedback and constructive suggestions have been invaluable in improving the quality of our work. We have carefully addressed all your recommendations, and the corresponding revisions are highlighted in blue within the revised manuscript.
We deeply appreciate your significant contributions throughout the review process and look forward to potential future discussions. Once again, thank you for your insightful and invaluable input.

Warm regards,

The Authors

---

### Author Response · Authors · 2024-12-04
**Summary for the Paper and Discussion**

Dear Esteemed Area chair and Reviewers,

We extend our sincere appreciation to the reviewers for their invaluable comments during the rebuttal process. Their insightful feedback has significantly bolstered the substance and clarity of our paper. In what follows, we will summarize the key contributions of our work and highlight the main points of our discussion with the reviewers.

****

**Main Contributions of Our Work**

**1) We proposed Logic-Logit, a novel rule-based interpretable choice model.**

In many choice modeling tasks, particularly those requiring decision transparency, such as in medical scenarios, it is crucial to understand the logic behind the decision-maker’s choices. We introduce rule-based utility value to discrete choice models, using OR-of-ANDs rules to depict decision makers' choice pattern.

**2) We creatively integrated the rule learning with columns generation to the Frank-Wolfe algorithm.**

We employ a column generation algorithm, integrating BFS (Breadth-First Search) and DFS (Depth-First Search) strategies, to tackle the intricate task of extracting decision logic rules from choice behaviors. By combining it with the Frank-Wolfe algorithm, our model efficiently learns a nuanced distribution of rule-based preference types.

**3) Our model outperforms baseline models in both interpretability and accuracy**

Our empirical evaluation on both synthetic datasets and real-world data from commercial and healthcare domains shows that the Logic-Logit model excels in interpretability and accuracy compared to baseline models. It effectively bridges the gap between identifying heterogeneity and clarifying underlying decision-making logic.

****

**Key Points of Discussion**

**The Generation of Predicates**

Some reviewers have expressed concerns about how we predefined the predicates. In response, we have detailed the generation process for both discrete and continuous features in Appendix A of the revised document. However, it is important to note that predefining predicates is not the central focus of our paper. The method used in our real dataset experiments is simply a general, straightforward example to illustrate the algorithm’s functionality. Predicates are defined as mappings from original features to binary {0, 1} features. Any model or algorithm that generates binary outputs can serve as a predicate generator, significantly enhancing the model’s flexibility to handle diverse data types, such as text data, as emphasized in our discussion with reviewer 7vMh.

**The Details of the Algorithm**

Some reviewers have expressed confusion about the specifics of our algorithm, particularly regarding how we employ BFS and DFS strategies to enhance the rule-searching process. In response, we have included detailed pseudo-codes in Appendix A to demonstrate how the algorithm executes the Frank-Wolfe step using column generation. Additionally, we have provided further formulations to elucidate the computation of rule potential. We are confident that this additional information will greatly clarify the rule-searching strategy for readers.

**The computational complexity and model capacity**

In response to concerns about computational complexity and model capacity, we clarify that the intensive computation is limited to the training phase due to the iterative nature of column generation. Practical stopping criteria, such as iteration limits or improvement thresholds, balance solution quality and computational time. For model capacity, our rule-based approach resembles basis discovery, incrementally identifying simple patterns and combining them to form complex rules. Starting with a large predicate set and scaling up gradually, the model avoids complexity and progressively captures intricate patterns through refinement.

**The scalability when facing large-scale dataset**

Some reviewers have expressed concerns about the model's performance on large datasets. We have supplemented experiments on a larger Expedia dataset, providing the hyperparameters that need to be adjusted for the model to adapt to larger datasets. Additionally, we conducted supplementary sensitivity analysis experiments for key hyperparameters.

**Promising Future Directions**

Discussions with our reviewers have inspired us to explore several promising future directions for extending our model, such as integrating it with graph-based methods, tuning LLM, and application in RLHF. We are particularly grateful to reviewer brV8 for suggesting a more refined approach to learning the mixed multinomial logit model, which offers better provable theoretical guarantees. Although this refined Frank-Wolfe algorithm heavily relies on panel data, as we highlighted in our discussions, combining our model with this strategy in the future will open up a broader range of applications.

---

### Meta-Review · Area_Chair_MVz6 · 2024-12-23

**Metareview:**

Summary
=======
The paper introduces Logic-Logit, a interpretable choice model that uses OR-of-ANDs logical rules to predict human choices. The key innovation is combining rule-based modeling with optimization techniques (Frank-Wolfe algorithm and column generation) to extract interpretable decision rules. The authors claim their approach achieves strong predictive performance while maintaining interpretability, demonstrated through experiments on synthetic data, Expedia Hotel dataset, and MIMIC-IV healthcare data.

Strengths
=======
* The method of choice modelling with interpretable rules is interesting
* The experimental evaluation is robust
* The presentation is generally clear and well-organized, supporting reproducibility.

Weaknesses
==========
* The computational complexity with larger datasets and feature spaces is not adequately addressed.
* Key details about hyperparameters, convergence conditions, and the exact rule search strategy are missing.
* The relationship to mixed-logit models and consider-then-choose models is not explored, leaving questions about the model's theoretical properties.
* There's insufficient analysis of how various parameters (number of rules, rule lengths, pruning thresholds) affect model performance.
* There is a rich literature on efficient tree-based methods to generate conjunctive rules, followed by (sparse) predicate selection. This should be evaluated against as well.

Reasons for decision
================
While the paper presents potentially valuable approach to interpretable choice modeling, the significant gaps in methodology, theoretical analysis, and scalability discussion need to be addressed.

**Additional Comments On Reviewer Discussion:**

The authors provided good responses and made improvements to the paper. The main concerns about scalability and implementation details were addressed through additional experiments and documentation, resulting in several increase in score. Some theoretical limitations remain, however.

---

### Decision · Program_Chairs · 2025-01-22

Accept (Poster)